# Colonizing multidrug-resistant bacteria and the longitudinal evolution of the intestinal microbiome after liver transplantation

Medini K. Annavajhala [1,2], Angela Gomez-Simmonds[1], Nenad Macesic[1,3], Sean B. Sullivan [1,2], Anna Kress[1], Sabrina D. Khan [1,2], Marla J. Giddins [1,2], Stephania Stump [1,2], Grace I. Kim [4], Ryan Narain [4], Elizabeth C. Verna[4] & Anne-Catrin Uhlemann [1,2]*

Infections by multidrug-resistant bacteria (MDRB) remain a leading cause of morbidity and mortality after liver transplantation (LT). Gut dysbiosis characteristic of end-stage liver disease may predispose patients to intestinal MDRB colonization and infection, in turn exacerbating dysbiosis. However, relationships between MDRB colonization and dysbiosis after LT remain unclear. We prospectively recruited 177 adult patients undergoing LT at a single tertiary care center. 16 S V3-V4 rRNA sequencing was performed on 723 fecal samples collected pre-LT and periodically until one-year post-LT to test whether MDRB colonization was associated with decreased microbiome diversity. In multivariate linear mixed-effect models, MDRB colonization predicts reduced Shannon α-diversity, after controlling for underlying liver disease, antibiotic exposures, and clinical complications. Importantly, pre-LT microbial markers predict subsequent colonization by MDRB. Our results suggest MDRB colonization as a major, previously unrecognized, marker of persistent dysbiosis. Therapeutic approaches accounting for microbial and clinical factors are needed to address post-transplant microbiome health.

[1] Department of Medicine, Division of Infectious Diseases, Columbia University Irving Medical Center, 630W 168th Street, New York, NY 10032, USA. [2] Microbiome and Pathogen Genomics Core, Columbia University Irving Medical Center, 630W 168th Street, New York, NY 10032, USA. [3] Central Clinical School, Monash University, Melbourne, Victoria, Australia. [4] Division of Digestive and Liver Diseases, Center for Liver Disease and Transplantation, Columbia University Irving Medical Center, NY Presbyterian Hospital, New York, NY, USA. *email: au2110@cumc.columbia.edu

Liver transplantation (LT) provides the only cure for chronic cirrhosis and end-stage liver disease[1]. However, infections due to multidrug-resistant bacteria (MDRB) represent an important cause of morbidity and mortality after solid organ transplantation[2]. Patients after LT are disproportionally affected by MDRB[3]. We and others have demonstrated a critical role for intestinal MDRB colonization prior to the development of infection[4,5]. MDRB infections, in turn, significantly impact post-LT survival[4–6], making colonization by MDRB a critical factor in the transplant-related clinical course. Recipients of LT receive frequent and sustained antibiotic regimens and routinely experience adverse clinical events after transplantation[4,7]. Antibiotic exposure has been well-established as an important factor in the emergence of MDRB[8–13], and even single antibiotic doses can negatively impact gut microbiome diversity[14,15]. Moreover, changes in gut microbiota composition and function contribute to progression of chronic liver disease and cirrhosis pre-transplant[16–18]. This suggests important yet complex interactions between MDRB, antibiotic exposure, and the gut microbiome, which directly impact pre-LT disease severity and post-transplant clinical outcomes.

The interdependence of the liver–gut axis confers an important role for gut microbiota and microbial products in healthy liver function, primarily through the regulation of lipid, choline, bile acid, and other metabolic pathways[19]. Conversely, increased gut permeability[20–22] along with quantitative and qualitative shifts in bacterial load and microbiome community structure[16,18,23,24], referred to as dysbiosis, have been associated with the progression of liver disease. Increases in pathobionts, such as Enterobacteriaceae and the corresponding decreases in protective microbiota, including Bacteroidaceae and Lachnospiraceae, have been reported in patients with liver disease, with or without cirrhosis[16,17,24–30]. Dysbiosis typically represents a significant perturbation in the number of taxa and their distribution within a sample (α-diversity) and/or relative similarity of overall microbial community composition across samples (β-diversity). Increased diversity represents a healthy gut[31], as a varied commensal community may protect against overgrowth of pathogenic bacteria and help modulate the innate immune response at homeostasis[32,33]. Mounting evidence is available regarding the association between low gut diversity and disease or poor clinical outcomes[34].

Although dysbiosis of the intestinal microbiome is dependent upon liver disease etiology and severity[25,27–30], the interplay between LT and dynamics of intestinal microbiota is complex and has rarely been studied. High model for end-stage liver disease (MELD) and Child–Turcotte–Pugh (CTP) scores were associated with reduced microbial diversity pre LT[16,35], but the impact of disease severity on the post-LT microbiome has not been reported. Previous reports with limited sample sizes found that microbial diversity decreased during the peri-operative period before subsequently increasing, and showed that reduced post-LT diversity was associated with acute cellular rejection and bloodstream infections[26,35,36]. However, important gaps in knowledge remain regarding associations between dysbiosis, MDRB colonization, and antibiotic use before and after LT, especially within the context of pre-transplant liver disease.

Here, we interrogate the complex relationship hypothesized between MDRB colonization and the gut microbiome after LT. Due to the likely importance of relevant clinical factors, including antibiotic exposure, liver disease, and post-transplant complications, we construct robust multivariate models assessing the ability of MDRB colonization to predict gut microbiome diversity before and after LT. Our results show that colonization by MDRB predicts reduced Shannon α-diversity throughout the post-transplant period, after controlling for the underlying liver etiology and disease severity, antibiotic exposures, and post-transplant clinical complications. Moreover, this relationship between colonizing MDRB, clinical factors, and gut dysbiosis is temporal, with specific MDRB, antibiotic classes, and post-LT complications significantly impacting microbiome diversity at different post-transplant phases. Importantly, we identify microbial markers present before LT, which predict subsequent MDRB colonization. Taken together, our results suggest MDRB colonization as a major, previously unrecognized, marker of persistent dysbiosis with clinical implications for microbiome-informed therapeutic approaches after not only LT but also other solid organ transplantations.

## Results

**Altered LT microbiome during colonization by MDRB**. We enrolled 195 patients pre-transplant, of whom 177 completed 1-year post-transplant follow-up (Supplementary Fig. 1). Patient characteristics are summarized in Table 1. The majority of patients were male (60%), and the median age was 60 years. Hepatitis C virus (HCV) was the most common reason for transplant ($n = 71$, 40%), followed by nonalcoholic fatty liver disease (NAFLD; $n = 30$, 17%) and alcohol-related liver disease (ARLD; $n = 19$, 11%), and 69 patients (39%) had concurrent hepatocellular carcinoma (HCC). The median MELD was 18 (IQR 13–24), and median CTP score was 9 (IQR 7–11) at the time of LT. A high proportion of patients (65%) developed intestinal MDRB colonization at least once over the 1-year study period. This included colonization by carbapenem-resistant Enterobacteriaceae (CRE, $n = 31$, 18%), Enterobacteriaceae resistant to third-generation cephalosporins (Ceph-RE, $n = 83$, 47%), and/or vancomycin-resistant enterococci (VRE, $n = 77$, 44%).

We first analyzed within-sample α-diversity, which incorporates the richness (number of taxa) and evenness (how these taxa are distributed) within the community. In unadjusted linear mixed-effect regression (LME) analysis, CRE, VRE, and Ceph-RE colonization were each associated with reduction in both Shannon and Chao α-diversity across all time points (LME Shannon $p < 0.0001$; Chao $p < 0.01$, Supplementary Data 1). VRE-colonized (culture-positive) compared with VRE-negative samples showed the most drastic reduction in both diversity indices (mean ($\pm$ standard deviation) Shannon of $2.32 \pm 0.86$ vs. $3.08 \pm 0.73$ and Chao of $189 \pm 73$ vs. $250 \pm 75$, respectively).

Microbiome β-diversity compares microbial communities between samples by generating a dissimilarity matrix, with pairwise distances calculated for every pair of samples. The permutational multivariate analysis of variance (PERMANOVA) test can then be used to assess whether the overall microbiome community structure (presence and relatedness of taxa across samples) is significantly different across two groups based on the β-diversity distance matrix. In our cohort, UniFrac β-diversity differed significantly in samples collected during colonization by CRE, VRE, and Ceph-RE, and in any MDRB-colonized compared with noncolonized samples (each PERMANOVA $P = 0.001$, Fig. 1, Table 4). Patients colonized with MDRB within 1 year post-LT also had significantly altered β-diversity across all time points (PERMANOVA $P = 0.001$, Table 4).

**Antibiotic exposure and gut microbial diversity**. There were remarkably high levels of antibiotic exposure in the study cohort. From 6 months pre-LT to 1 year post-LT, the median number of days spent on inpatient antibiotic courses was 16 (IQR 8–33) (Table 1). Notably, 21 patients (12%) received >60 days of antibacterial treatment. Furthermore, the vast majority (96%) of patients received courses of multiple antibacterial agents

**Table 1 Description of the study cohort**

| Clinical characteristic | n = 177 |
|---|---|
| *Demographics and anthropometrics* | |
| Age (years) (median, IQR) | 60 (54–65) |
| Sex, male | 107 (60%) |
| UNOS race/ethnicity—White | 110 (62%) |
| Asian | 20 (11%) |
| Black/African American | 15 (8%) |
| Hispanic/Latino | 32 (25%) |
| *Primary underlying etiology* | |
| Hepatitis C infection (HCV) | 71 (40%) |
| Nonalcoholic fatty liver disease (NAFLD) | 30 (17%) |
| Alcohol (ARLD) | 19 (11%) |
| Biliary-related (PSC/PBC/BC/other) | 18 (10%) |
| Autoimmune hepatitis (AIH) | 13 (7%) |
| Hepatitis B infection (HBV) | 10 (6%) |
| Polycystic liver/kidney disease (PCLD) | 6 (3%) |
| Cryptogenic liver disease (CLD) | 3 (2%) |
| Other | 7 (4%) |
| *Comorbidities* | |
| Co-existing hepatocellular carcinoma (HCC) | 69 (39%) |
| Alpha-1 antitrypsin deficiency | 5 (3%) |
| *Liver disease severity* | |
| MELD score at LT (median, IQR) | 18 (13–24) |
| CTP score at LT (median, IQR) | 9 (7–11) |
| *Transplant characteristics* | |
| Living donor | 30 (17%) |
| Cold ischemic time (minutes) (median, IQR) | 326 (244–458) |
| PRBC transfusion required | 137 (77%) |
| PRBC units transfused (median, IQR) | 6 (3–10) |
| *Post-operative complications* | |
| Bleeding (≤ 7 days post-LT) | 34 (19%) |
| Biliary stricture (≤ 365 days post-LT) | 26 (15%) |
| Biliary leak (≤ 365 days post-LT) | 17 (10%) |
| *Outcomes* | |
| Readmission to hospital | 93 (53%) |
| Readmission to ICU | 50 (28%) |
| 365 days post-LT all-cause mortality | 6 (3%) |
| *MDRB and antibiotic usage* | |
| Peri-operative adjustments to antibiotic regimen | 65 (37%) |
| Inpatient days on antibacterial treatment courses (−0.5 to 1 yr post-LT) | 16 (8–33) |
| Number of antibacterial agents received (−0.5 to 1 yr post-LT) | 5 (3–7) |
| *MDRB colonization (≤ 365 days post-LT)* | 115 (65%) |
| CRE colonization | 31 (18%) |
| Ceph-RE colonization | 83 (47%) |
| VRE colonization | 77 (44%) |

*UNOS* United Network for Organ Sharing, *PSC* primary sclerosing cholangitis, *PBC* primary biliary cirrhosis, *BC* biliary cholangitis, *MELD* model for end-stage liver disease (2016 Organ Procurement and Transplantation Network guidelines), *CTP* Child–Turcotte–Pugh, *LT* liver transplant, *PRBC* packed red blood cells, *MDRB* multidrug-resistant bacteria, *CRE* carbapenem-resistant Enterobacteriaceae, *Ceph-RE* third-generation cephalosporin-resistant Enterobacteriaceae, *VRE* vancomycin-resistant enterococci

analyses, exposures to group 2 β-lactams (including third-generation cephalosporins and first-generation β-lactamase inhibitor combinations (e.g., piperacillin–tazobactam)), glyco-/lipo-peptides (e.g., vancomycin and daptomycin), and carbapenems within 14 days were associated with significantly lower α-diversity both pre- and post-LT (Tables 2, 3). Fluoroquinolone exposure was also associated with reduced α-diversity up to 6 months post-LT. This association was not observed in the later post-LT period. Exposure to penicillins and first- and second-generation cephalosporins (denoted as group 1 β-lactams) did not have a significant association with α-diversity in either direction (Supplementary Data 2).

**Clinical predictors of pre-transplant gut microbiome diversity.** In order to construct a robust multivariate model describing the roles of MDRB and antibiotic exposure before transplantation, we assessed associations between multiple clinical confounders relevant to end-stage liver disease and the pre-transplant microbiome by using linear regression models (LM).

Patient demographics (age, race, ethnicity, and sex) were not significantly associated with pre-transplant α-diversity (Supplementary Data 1). However, the underlying liver disease etiology was significantly associated with pre-LT gut microbiome diversity (LM Shannon $P < 0.01$; Chao $P < 0.01$; Table 2). In particular, ARLD patients had significantly lower Shannon α-diversity than those with other etiologies (Figs. 2a, 3, Table 2). In contrast, patients with HCC had significantly higher Shannon α-diversity than those without cancer (LM $P < 0.05$). Patients with ARLD also had distinct pre-LT microbiome community structure compared with other diagnoses (UniFrac PERMANOVA $P = 0.014$) (Fig. 2b, Table 4).

ARLD patients had significantly lower *Faecalibacterium prausnitzii* and were enriched in *Streptococcus* and *Lactobacillus* OTUs, including *Lactobacillus zeae,* in differential abundance testing as identified by *DESeq2* (FDR-adjusted $P < 0.05$) (Supplementary Data 3). Given the potential for high false-positive rates by using *DESeq2*[37], however, we also used the Analysis of Composition of Microbiomes (ANCOM) method to refine our identification of bacterial taxa enriched across group comparisons[38]. By using ANCOM, we confirmed that several *Streptococcus* and *Lactobacillus* OTUs, and *L. zeae*, were differentially enriched pre-LT in patients with ARLD (W-statistic > 0.7) (Supplementary Data 4).

Patients with severe liver disease (MELD > 18 or CTP class C) had significantly lower pre-LT bacterial α-diversity (Fig. 2c, e, Table 2; Supplementary Data 1). High-MELD patients harbored distinct pre-LT microbiome structures compared with low-MELD patients (UniFrac PERMANOVA $P = 0.001$) (Fig. 2d, Table 4). *DESeq2* identified significant enrichment in *Enterococcus casseliflavus, Veillonella dispar,* and *L. zeae* in high-MELD patients, and differential enrichment of *F. prausnitzii, Dialister, Bifidobacterium bifidum,* and *Bifidobacterium adolescentis,* and multiple *Bacteroides* OTUs in low-MELD patients (FDR-adjusted $P < 0.05$) (Supplementary Data 5). ANCOM analysis identified significant enrichment of *Enterococcus* and *Lactobacillus zeae* in high-MELD patients and of taxa traditionally associated with a healthier gut (Lachnospiraceae, *F. prausnitzii, Collinsella aerofaciens*) in low-MELD patients (W > 0.7; Supplementary Data 6).

Patients with CTP class C also had distinct microbial communities compared with CTP class A patients, while CTP class B communities overlapped with both class A and C (UniFrac PERMANOVA $P = 0.001$; Fig. 2f, Table 4). CTP class C was associated with significant enrichment in several *Enterococcus* and *Lactobacillus* OTUs, while class A was characterized by enrichment in *F. prausnitzii, Bifidobacterium,* similarly to patients with low MELD (*DESeq2,* FDR-adjusted $P < 0.05$)

throughout the study period (median five agents, IQR 3–7), with 12 patients (7%) receiving at least 10 different antibacterials.

Antibiotic exposure within 14 days prior to sample acquisition was associated with significantly decreased gut microbial diversity at any phase pre- and post-LT (LME Shannon $p < 0.0001$; Chao $p < 0.0001$) in univariate mixed-effect regression (Supplementary Data 1). Gut community structure was also significantly altered in samples collected within 14 days of treatment with any antibiotic, regardless of drug used or pre-/post-LT sampling time point, compared with samples not collected within this window (UniFrac PERMANOVA $P = 0.001$) (Table 4). Importantly, we observed that the effect of antibiotic exposure on gut α-diversity was dependent on antibiotic class. In univariate regression

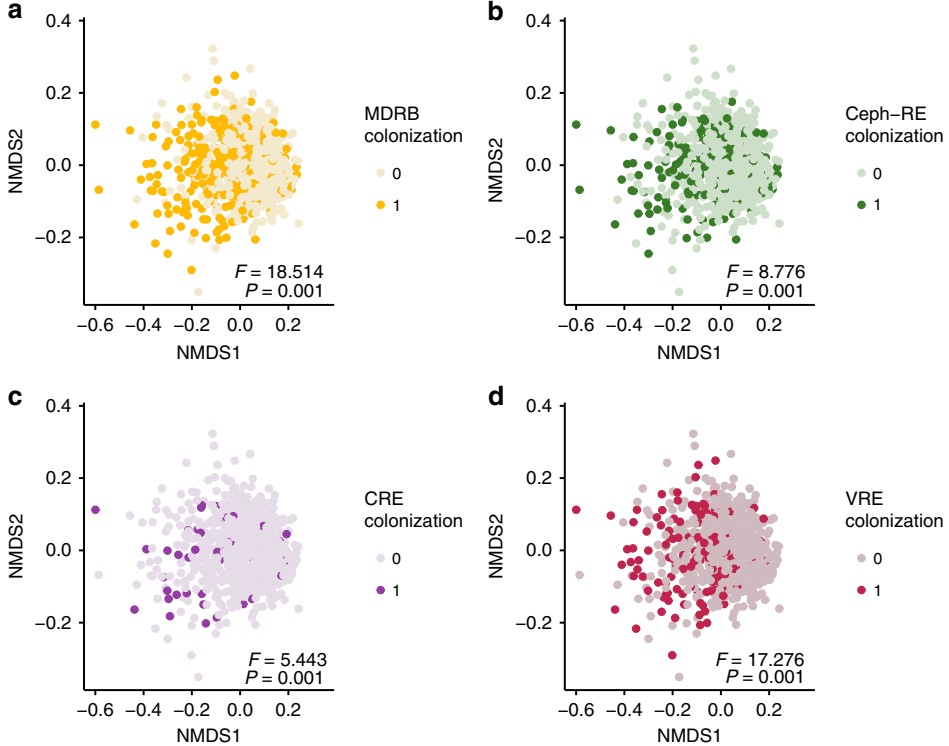

**Fig. 1** Significant clustering of samples collected during colonization by multidrug-resistant bacteria (MDRB). Nonmetric multidimensional scaling (NMDS) plots were constructed by using UniFrac β-diversity. Each point represents a single sample in the full data set ($n = 703$). The difference in microbiome community structure was compared during versus not during colonization by (**a**) any MDRB (dark yellow; colonized $n = 254$ vs. noncolonized $n = 449$), **b** vancomycin-resistant enterococci (VRE, dark pink; colonized $n = 142$ vs. noncolonized $n = 561$), **c** Enterobacteriaceae resistant to third-generation cephalosporins (Ceph-RE, dark green; colonized $n = 162$ vs. noncolonized $n = 541$), and **d** carbapenem-resistant Enterobacteriaceae (CRE, dark purple; colonized $n = 46$ vs. noncolonized $n = 657$) as identified by using culture-based methods (see Methods, Supplementary Methods). PERMANOVA pseudo-F-statistic and P-value are shown. Source data are provided as a Source Data file

(Supplementary Data 7). Notably, the same *Enterococcus* and *F. prausnitzii* OTUs were enriched pre-LT in high-MELD and CTP class C and low-MELD and CTP class A patients, respectively, based on both *DESeq2* and ANCOM (Supplementary Data 5–8).

After adjusting for ARLD diagnosis, CTP score, and recent exposure to glyco-/lipo-peptides, VRE ($P < 0.1$) and CRE ($P < 0.01$) colonization at sampling were independent predictors of reduced pre-LT Shannon α-diversity in a multivariate LM model (overall $P < 0.0001$) (Table 2, Fig. 4).

**Distinct gut microbiome dynamics across post-operative phases.** We also observed distinct dynamics in gut microbiome α-diversity and β-diversity across three post-transplant phases: the peri-operative (weeks 1–3), early post-transplant (months 1–3), and late post-transplant (months 6–12) periods (Fig. 3). All patients displayed a significant shift in microbial community structure immediately after transplant compared with pre-LT, and from pre- and peri-LT to post-LT (months 1–12) regardless of diagnosis (UniFrac PERMANOVA $P = 0.001$) (Fig. 3, Table 4).

We used constrained linear mixed-effect (CLME) analysis to test for significant changes in α-diversity over time within each disease group (Supplementary Data 9, 10). Based on global bootstrap LRT testing through CLME, both Shannon and Chao α-diversity significantly differed across time points for most major diagnoses (autoimmune hepatitis (AIH), ARLD, biliary-related disease, and HCV; CLME $P < 0.05$). Only Chao α-diversity showed a significant longitudinal progression in patients with NAFLD (CLME Shannon $P = 0.140$; Chao $P < 0.05$), and hepatitis B virus (HBV) patients showed no significant temporal trend in

either α-diversity metric. Of note, for patients with ARLD, both Shannon and Chao α-diversity were the lowest pre-LT, while for all other diagnoses, week 1 post-LT represented the most significant reduction in α-diversity. More granular comparisons across time points by using CLME (individual θ and P-values) revealed interesting trends in α-diversity over time: for patients with AIH, biliary-related etiologies, and HCV, increases in α-diversity were significant ($P < 0.05$) primarily between week 3 to month 1 post-LT and month 3 to month 6 post-LT. This further justified our decision to analyze the early (months 1–3) and late (months 6–12) post-LT phases separately in subsequent analyses. In contrast, ARLD patients, likely due to low pre-LT diversity, had a drastic increase in Shannon and Chao α-diversity earlier in their post-LT course (pre-LT to week 2 post-LT).

We identified several clinical parameters that were significantly associated with peri-LT gut microbiome α-diversity and β-diversity in univariate regression analyses. Higher CTP score and MELD were both associated with decreased α-diversity (LME Shannon and Chao $P < 0.0001$) through the peri-LT period (Table 2). Bleeding within 7 days of transplant ($n = 34$, 19%; Table 1) and adjusted peri-operative antibiotic regimen were associated with significantly lower peri-LT α-diversity (Table 2). Patients requiring peri-operative antibiotic regimen adjustment additionally had altered β-diversity (PERMANOVA $P = 0.001$) throughout the peri-operative period (Table 4). Patients with ARLD and HBV had distinct β-, but not Shannon α-diversity, from all other patients during the peri-LT time period (Table 2, Table 4; Supplementary Data 1). Compared with pre-LT samples, the peri-LT microbiome was significantly enriched (*DESeq2*, FDR-adjusted $P < 0.05$) in multiple OTUs within Clostridiales, *Streptococcus*, and *Enterococcus* (including *E.*

**Table 2 Multivariate linear mixed-effect regression models assessing clinical and microbial predictors associated with gut Shannon α-diversity before and immediately after liver transplantation**

| Predictor | Univariate coefficient | Univariate Pa | Multivariate coefficient (95% CI) | Multivariate Pa,b |
|---|---|---|---|---|
| *Pre-LT* | | | | |
| Liver disease—ARLD | | Reference | −0.72 (−1.35, −0.09) | <0.05 |
| AIH | 1.30 | <0.01 | | |
| BILIARY | 0.91 | 0.11 | | |
| HBV | 1.39 | <0.01 | | |
| HCV | 1.45 | <0.001 | | |
| NAFLD | 1.48 | <0.001 | | |
| Comorbidity—HCC | 0.42 | <0.05 | | |
| MELD (>18 vs. ≤18) | −0.99 | <0.0001 | | |
| CTP score | −0.22 | <0.0001 | −0.14 (−0.21, −0.07) | <0.0001 |
| CRE colonization | −2.28 | <0.001 | −1.48 (−2.54, −0.42) | <0.01 |
| VRE colonization | −1.03 | <0.001 | −0.43 (−0.91, −0.05) | 0.08 |
| Ceph-RE colonization | −0.46 | 0.1 | | |
| β-lactam IIc | −0.54 | 0.07 | | |
| Glyco-/lipo-peptidec | −1.30 | <0.001 | −0.61 (−1.21, −0.01) | <0.05 |
| Carbapenemc | −0.90 | <0.05 | | |
| Fluoroquinolonec | −0.98 | <0.001 | | |
| *Peri-LT (weeks 1–3)* | | | | |
| Comorbidity—HCC | 0.31 | <0.05 | | |
| MELD (>18 vs. ≤18) | 0.67 | <0.0001 | | |
| CTP score | −0.17 | <0.0001 | −0.11 (−0.16, −0.07) | <0.0001 |
| PRBC transfusion (units) | −0.02 | <0.01 | | |
| Post-operative bleedingd | −0.69 | <0.001 | −0.45 (−0.75, −0.15) | <0.01 |
| Peri-operative antibiotic adjustment | −0.67 | <0.0001 | −0.50 (−0.73, −0.23) | <0.0001 |
| CRE colonization | −0.59 | <0.01 | | |
| VRE colonization | −0.41 | <0.001 | | |
| Ceph-RE colonization | −0.42 | <0.01 | −0.29 (−0.52, −0.07) | <0.05 |
| β-lactam IIc | −0.26 | <0.01 | −0.24 (−0.42, −0.05) | <0.05 |
| β-lactam IIIc | −0.76 | <0.05 | | |
| Glyco-/lipo-peptidec | −0.33 | <0.05 | | |
| Carbapenemc | −0.77 | <0.01 | | |
| Fluoroquinolonec | −0.65 | <0.001 | −0.55 (−0.89, −0.21) | <0.01 |

aUni- and multivariate associations were assessed by using linear (pre-LT) or linear mixed-effect (peri-LT) regression models. Predictors with univariate $P \leq 0.1$ were considered for inclusion in multivariate modeling
bAll multivariate models had overall (global) $P < 0.0001$
cAntibiotic exposure ≤ 14 days before sample; see Supplementary Methods for coding of antibiotic classes
dWithin 7 days of LT

*casseliflavus*), as well as *L. zeae*, adjusted for primary liver disease etiology (Supplementary Data 11).

Regardless of liver disease etiology, the early and late post-LT phases were characterized by distinct microbiome communities (UniFrac PERMANOVA $P = 0.001$; Fig. 3). Differential abundance testing by using *DESeq2*, controlling for liver disease, revealed that early post-LT samples were enriched in Clostridiales, and had significantly lower levels of *Lactobacillus* and *Bacteroides* compared with peri-LT samples (Supplementary Data 12). Late post-LT samples were, in turn, significantly enriched in *Ruminococcus bromii, Butyricicoccus pullicaecorum,* and *Blautia producta* compared with early post-LT, and had significantly lower levels of *Enterococcus* and *Streptococcus* (Supplementary Data 13). We also used ANCOM to analyze the differential abundance of gut microbiota across transplant phases while adjusting for repeated sampling and liver disease etiology. As with *DESeq2*, both pathogenic taxa—e.g., *Enterococcus* (including *E. casseliflavus*), *Lactobacillus* (including *L. zeae*), and *Streptococcus*—and protective taxa—e.g., *F. prausnitzii* and Ruminococcaceae—were differentially abundant across pre-LT to post-LT phases (Supplementary Data 14).

Based on the patterns observed in both α-diversity and β-diversity, as well as differential abundance of specific microbiota, we used separate regression models to assess modulation of post-LT gut microbiome diversity during the early (months 1–3) and late (months 6–12) post-transplant periods (Table 3). As during

peri-LT, liver disease severity, peri-operative antibiotic adjustment, and post-operative bleeding were associated with lower α-diversity during early and late post-LT in unadjusted mixed-effect regression. We also found significant univariate associations between postoperative biliary complications, and readmission to the hospital and/or intensive care unit (ICU), and early and late post-LT α-diversity.

In multivariate models, MDRB colonization was significantly associated with reduced α-diversity throughout the post-LT course (Table 3), although individual MDRB were prominent during different post-transplant phases. Colonization by Ceph-RE was an independent predictor of reduced peri-LT Shannon α-diversity ($P < 0.05$) after adjusting for exposure to group 2 β-lactams or fluoroquinolones, high CTP score, peri-operative antibiotic regimen adjustment, and postoperative bleeding (multivariate LME model $P < 0.0001$) (Fig. 4). Early post-LT Shannon α-diversity was negatively associated with colonization by CRE ($P < 0.01$) after adjusting for exposure to group 2 β-lactams, glyco-/lipo-peptides, or fluoroquinolones, transplant MELD, peri-operative antibiotic adjustment, and bile leak within 1 year post-LT (multivariate LME $P < 0.0001$) (Fig. 4). In contrast, VRE colonization ($P < 0.0001$) was significantly associated with late post-LT Shannon α-diversity in a multivariate model after adjusting for MELD, peri-operative antibiotic regimen, bile leak, or biliary stricture within 1 year post-LT, and exposure to group 2 β-lactams (multivariate LME $P < 0.0001$; Fig. 4).

**Table 3 Multivariate linear mixed-effect regression models assessing clinical and microbial predictors associated with gut microbiome α-diversity after liver transplantation**

| Predictor | Univariate coefficient | Univariate P[a] | Multivariate coefficient (95% CI) | Multivariate P[a,b] |
|---|---|---|---|---|
| *Early post-LT (months 1–3)* | | | | |
| Comorbidity—HCC | 0.30 | <0.05 | | |
| MELD (>18 vs. ≤18) | −0.34 | <0.001 | −0.34 (−0.51, −0.16) | <0.001 |
| CTP score | −0.07 | <0.01 | | |
| Post-operative bleeding[c] | −0.43 | <0.01 | | |
| Bile leak[d] | −0.73 | <0.0001 | −0.49 (−0.77, −0.22) | <0.001 |
| Biliary stricture[d] | −0.45 | <0.01 | | |
| Re-hospitalization[d] | −0.25 | <0.05 | | |
| Readmission to ICU[d] | −0.37 | <0.01 | | |
| Peri-operative antibiotic adjustment | −0.27 | <0.05 | −0.17 (−0.36, 0.01) | 0.07 |
| CRE colonization | −0.58 | <0.01 | −0.46 (−0.78, −0.14) | <0.01 |
| VRE colonization | −0.53 | <0.0001 | | |
| Ceph-RE colonization | −0.23 | 0.07 | | |
| β-lactam II[e] | −0.84 | <0.0001 | −0.41 (−0.68, −0.14) | <0.01 |
| β-lactam III[e] | −0.6 | <0.05 | | |
| Glyco-/lipo-peptide[e] | −0.96 | <0.0001 | −0.39 (−0.7, −0.08) | <0.05 |
| Carbapenem[e] | −0.72 | <0.001 | | |
| Fluoroquinolone[e] | −0.73 | <0.0001 | −0.46 (−0.75, −0.17) | <0.01 |
| *Late post-LT (months 6–12)* | | | | |
| Comorbidity—HCC | 0.23 | <0.05 | | |
| MELD (>18 vs. ≤18) | −0.31 | <0.01 | −0.21 (−0.39, −0.03) | <0.05 |
| CTP score | −0.05 | <0.05 | | |
| Post-operative bleeding[c] | −0.29 | <0.05 | | |
| Bile leak[d] | −0.41 | <0.05 | −0.33 (−0.64, −0.03) | <0.05 |
| Biliary stricture[d] | −0.52 | <0.001 | −0.33 (−0.58, −0.07) | <0.05 |
| Re-hospitalization[d] | −0.25 | <0.05 | | |
| Peri-operative antibiotic adjustment | −0.19 | 0.09 | −0.18 (−0.37, −0.01) | 0.07 |
| CRE colonization | −0.36 | <0.05 | | |
| VRE colonization | −0.70 | <0.0001 | −0.53 (−0.78, −0.29) | <0.0001 |
| Ceph-RE colonization | −0.20 | 0.05 | | |
| β-lactam II[e] | −0.46 | <0.001 | −0.29 (−0.54, −0.03) | <0.05 |
| Glyco-/lipo-peptide[e] | −0.51 | <0.001 | | |
| Carbapenem[e] | −0.55 | <0.05 | | |

[a]Univariate and multivariate associations were assessed by using linear mixed-effect regression models. Predictors with univariate $P \leq 0.1$ were considered for inclusion in multivariate modeling
[b]All multivariate models had overall (global) $P < 0.0001$
[c]Within 7 days of LT
[d]Within 1 year of LT
[e]Antibiotic exposure ≤ 14 days before sample; see Supplementary Methods for coding of antibiotic classes

**Microbial signatures associated with MDRB.** Given the significant association of MDRB colonization with α-diversity throughout the pre- and post-transplant period, we assessed possible predictors of post-transplant MDRB colonization across all time points. Notably, low pre-transplant Shannon α-diversity was associated with subsequent post-LT colonization by any MDRB (LM $P < 0.05$), including colonization specifically by CRE (LM $P < 0.05$), but not Ceph-RE or VRE. Furthermore, pre-LT β-diversity was significantly altered in patients who developed MDRB colonization at any point post-LT as compared with patients who were never colonized (PERMANOVA $P = 0.021$). Patients who never developed MDRB colonization were significantly enriched before LT in *F. prausnitzii*, *Bacteroides*, and *Bifidobacterium*, as well as *Parabacteroides distasonis*, *Prevotella copri*, and *Prevotella stercorea* (*DESeq2*, FDR-adjusted $P < 0.05$) (Supplementary Data 15). Both *DESeq2* and ANCOM identified the same *Enterococcus* OTU as enriched pre-LT in patients who developed MDRB within 1 year post-LT, and ANCOM identified both *Blautia obeum* and *Dorea formicigenerans* as potential pre-LT markers of subsequent MDRB colonization (Supplementary Data 16).

Overall, as expected based on culture results, samples collected during CRE colonization were differentially enriched in several Enterobacteriaceae OTUs, including *Klebsiella* and *Enterobacter cloacae*, through both *DESeq2* and ANCOM analyses

(Supplementary Data 17, 18). These samples also had significantly higher levels of *Streptococcus*, including *Streptococcus anginosus*. CRE-colonized samples harbored significantly reduced levels (*DESeq* FDR-adjusted $P < 0.05$, ANCOM $W > 0.7$) of *R. gnavus*, *F. prausnitzii*, and OTUs assigned to the *Bacteroides* and *Blautia* genera. Similarly, both *DESeq2* and ANCOM identified enrichment of several Enterobacteriaceae, including OTUs assigned to *Klebsiella* and *E. cloacae*, *Streptococcus*, and *Enterococcus* (including *E. casseliflavus*), and significantly lower levels of *Blautia*, *C. aerofaciens*, and several *Bacteroides* OTUs during colonization by Ceph-RE (Supplementary Data 19, 20). VRE colonization corresponded with a significant (*DESeq* FDR-adjusted $P < 0.05$, ANCOM $W > 0.7$) decrease in *F. prausnitzii*, multiple Lachnospiraceae, and *Blautia* (Supplementary Data 21, 22). Conversely, samples collected during VRE colonization were significantly enriched not only in *Enterococcus*, including *E. casseliflavus*, but also *L. zeae*, *Staphylococcus*, and *Klebsiella*.

Patients colonized with any MDRB within 1 year of transplant had significantly distinct β-diversity compared with those who were never colonized (PERMANOVA $P = 0.001$; Table 4). Importantly, these patients had increased levels of *Enterococcus* and *Klebsiella* and reduced levels of *Bacteroides*, *Faecalibacterium*, and *Lachnospira* not only during colonization but also throughout the entire study period (Supplementary Fig. 2). Differential

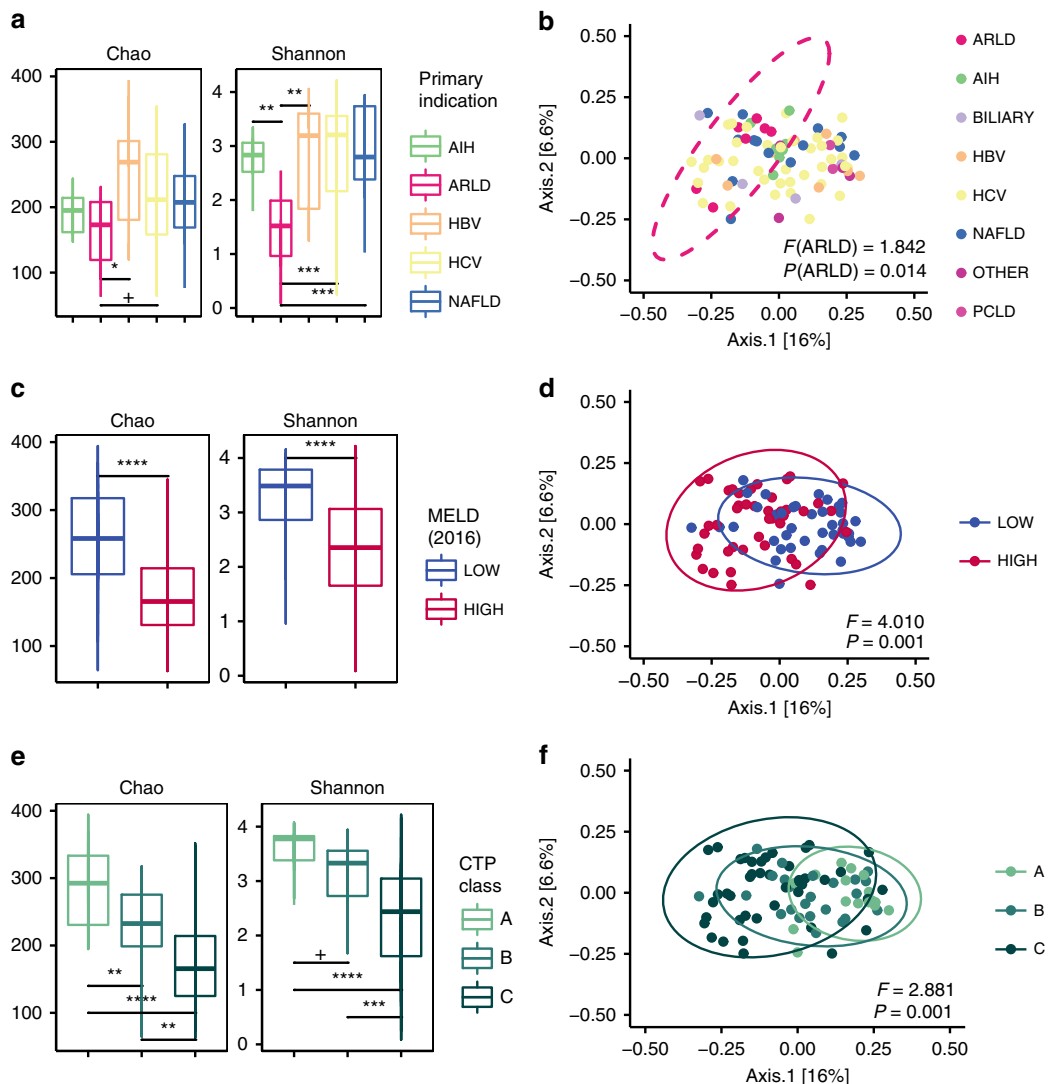

**Fig. 2** Pre-transplant α-diversity and β-diversity associated with liver disease etiology and severity. **a** Shannon and Chao α-diversity and **b** UniFrac β-diversity indices stratified by primary indication for LT (AIH $n = 7$; ARLD $n = 7$; HBV $n = 8$; HCV $n = 37$; NAFLD $n = 14$). **c**, **d** α-diversity and UniFrac β-diversity stratified by high vs. low MELD score at the time of LT (> 18 ($n = 43$) vs. ≤ 18 ($n = 40$)). **e**, **f** α-diversity and UniFrac β-diversity stratified by CTP class at the time of LT (**a** ($n = 16$) vs. **b** ($n = 24$) vs. **c** ($n = 43$)). **a**, **c**, **e** α-Diversity boxplots reflect median (horizontal center line), 25th and 75th percentile values (bottom and top bounds of boxes), and ranges (bottom and top of whiskers) for each category. Each panel shows *P*-values for univariate linear regression (see Table 2) as follows: + $P < 0.1$; *$P < 0.05$; **$P < 0.01$; ***$P < 0.001$; ****$P < 0.0001$. **b**, **d**, **f** PERMANOVA *P*-values and pseudo-F-statistic values calculated using UniFrac β-diversity distances are shown. LT liver transplant, AIH autoimmune hepatitis, ARLD alcohol-related liver disease, BILIARY biliary-related etiologies, HBV hepatitis B virus, HCV hepatitis C virus, NAFLD nonalcoholic fatty liver disease, PCLD polycystic liver/kidney disease, MELD model for end-stage liver disease, CTP Child–Turcotte–Pugh. Source data are provided as a Source Data file

abundance testing by using both *DESeq2* and ANCOM revealed that, across all time points, those who were colonized by MDRB at any point were significantly enriched in *Enterococcus* (including *E. casseliflavus*), *Klebsiella*, *E. cloacae*, as well as *L. zeae*, while those who never developed MDRB colonization were enriched in *Bacteroides*, *Blautia* (including *B. obeum*), *C. aerofaciens*, and multiple Lachnospiraceae and Ruminococcaceae OTUs (Supplementary Data 23, 24).

## Discussion

Liver cirrhosis has been associated with profound changes of the gut microbiota. Here, we demonstrate in a large prospective cohort of LT patients that severity and etiology of the underlying liver disease, along with antibiotic exposure and peri- and post-LT transplant-related complications,

significantly contribute to modulation of the gut microbiome well into the post-transplant period. Furthermore, our results indicate a strong, likely bidirectional, association between microbiome diversity and intestinal colonization with MDRB. Dense longitudinal sampling and rich MDRB-related data uniquely positioned our cohort to provide a granular view of microbial dynamics after LT and allowed for robust evaluation of critical pathogen-commensal, and specifically drug-resistant pathogen-commensal, relationships within the context of LT.

We identified microbial signatures of colonization by MDRB, and by CRE, VRE, and/or Ceph-RE specifically. Patients who developed MDRB colonization within 1 year post-LT had distinct β-diversity compared with those who never became colonized, with increased *Enterococcus* and *Klebsiella* and reduced levels of *Bacteroides* and *Lachnospira* throughout the entire study period. Importantly, pre-LT microbial β-diversity and levels of specific

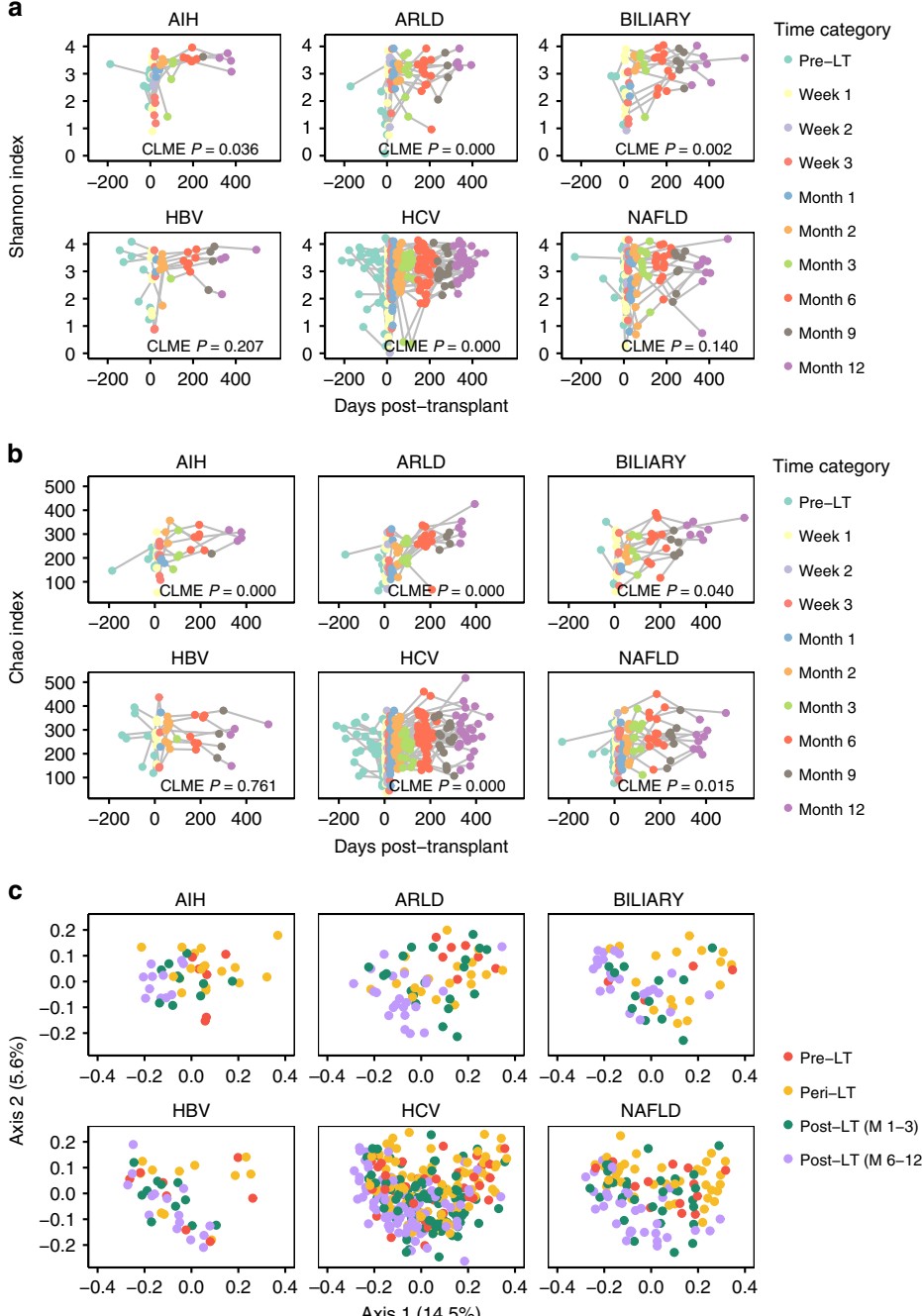

**Fig. 3** Progression of gut microbiome diversity before and after transplantation stratified by the underlying liver disease. Shannon (**a**) and Chao (**b**) α-diversity and UniFrac β-diversity (**c**) throughout the study period are shown. **a**, **b** AIH (13 patients, 45 samples), ARLD (19 patients, 69 samples), BILIARY (18 patients, 58 samples), HBV (10 patients, 45 samples), HCV (71 patients, 304 samples), and NAFLD (30 patients, 115 samples) were the most represented indications for LT in our cohort. These panels show Shannon (**a**) and Chao (**b**) α-diversity within each sample by the number of days post-LT of sampling for each disease group. Constrained linear mixed-effect (CLME) analysis was used to determine a global *P*-value describing the temporal trend in α-diversity within each disease group (see Supplementary Data 9 and Supplementary Data 10). **c** Changes in UniFrac β-diversity in the pre-, peri- (weeks 1–3), and early (months 1–3) and late (months 6–12) post-transplant periods are shown for patients with AIH (pre-LT $n = 7$; peri-LT $n = 17$; early post-LT $n = 10$; late post-LT $n = 11$), ARLD (pre-LT $n = 7$; peri-LT $n = 20$; early post-LT $n = 20$; late post-LT $n = 22$), BILIARY (pre-LT $n = 4$; peri-LT $n = 17$; early post-LT $n = 12$; late post-LT $n = 25$), HBV (pre-LT $n = 8$; peri-LT $n = 11$; early post-LT $n = 11$; late post-LT $n = 15$), HCV (pre-LT $n = 37$; peri-LT $n = 88$; early post-LT $n = 89$; late post-LT $n = 90$), and NAFLD (pre-LT $n = 14$; peri-LT $n = 38$; early post-LT $n = 31$; late post-LT $n = 32$). UniFrac β-diversity calculation and principal coordinates analysis (PCoA) were performed with the full dataset, and samples from each disease population were subsequently separated for visualization of temporal trends. Across all samples, the PCoA axis 1 explained 14.5% of observed variance, and PCoA axis 2 explained 5.6% of observed variance. LT liver transplant, AIH autoimmune hepatitis, ARLD alcohol-related liver disease, BILIARY biliary-related etiologies, HBV hepatitis B virus, HCV hepatitis C virus, NAFLD nonalcoholic fatty liver disease. Source data are provided as a Source Data file

**Table 4 Clinical and microbial predictors associated with gut microbiome UniFrac β-diversity before and after liver transplantation using PERMANOVA**

| Predictor | P | Pseudo F-statistic |
|---|---|---|
| *All time points* | | |
| CRE colonization[a] | 0.001 | 5.443 |
| VRE colonization[a] | 0.001 | 17.276 |
| Ceph-RE colonization[a] | 0.001 | 8.776 |
| Any MDRB colonization[a] | 0.001 | 18.514 |
| Any MDRB colonization[b] | 0.001 | 11.002 |
| Time category (pre-, peri-, and post-LT) | 0.001 | 8.488 |
| Antibacterial treatment within 14 days | 0.001 | 21.291 |
| *Pre-LT* | | |
| ARLD | 0.014 | 1.842 |
| MELD (>18 vs. ≤18) | 0.001 | 4.010 |
| CTP class | 0.001 | 2.881 |
| Any MDRB colonization[a] | 0.001 | 3.986 |
| Antibacterial treatment within 14 days | 0.001 | 3.485 |
| *Peri-LT (weeks 1–3)* | | |
| ARLD | 0.154 | 1.235 |
| HBV | 0.070 | 1.388 |
| Peri-operative antibiotic adjustment | 0.001 | 4.274 |
| Any MDRB colonization[a] | 0.001 | 6.559 |
| Antibacterial treatment within 14 days | 0.029 | 1.600 |
| *Early post-LT (months 1–3)* | | |
| Bile leak[b] | 0.002 | 2.623 |
| Biliary stricture[b] | 0.009 | 1.826 |
| Re-hospitalization[b] | 0.001 | 3.425 |
| Readmission to ICU[b] | 0.001 | 2.739 |
| Peri-operative antibiotic adjustment | 0.001 | 2.419 |
| Any MDRB colonization[b] | 0.001 | 3.959 |
| Antibacterial treatment within 14 days | 0.001 | 6.036 |
| *Late post-LT (months 6–12)* | | |
| Bile leak[b] | 0.049 | 1.434 |
| Biliary stricture[b] | 0.002 | 2.751 |
| Re-hospitalization[b] | 0.001 | 3.690 |
| Readmission to ICU[b] | 0.088 | 1.314 |
| Peri-operative antibiotic adjustment | 0.067 | 1.388 |
| Any MDRB colonization[a] | 0.001 | 3.124 |
| Antibacterial treatment within 14 days | 0.001 | 2.801 |

[a]At the time of sampling
[b]Within 1 year of LT

taxa were also significantly altered in patients who subsequently developed MDRB colonization. More specifically, we identified low pre-LT α-diversity as a significant marker for CRE colonization after transplantation. This suggests that gut microbial health prior to LT is a major contributor to post-transplant complications related to MDRB.

Our results significantly expand current understanding of the dynamic post-transplant evolution of the gut microbiome. In addition to colonization by MDRB, we established the long-term role of liver disease severity and etiology (i.e., ARLD), in terms of CTP score and MELD, on the gut microbiome throughout the pre-, peri-, and even late-post-LT periods. *E. casselflavius* and *L. zeae* enriched pre-LT in patients with ARLD, as well as those with high MELD and/or CTP, emerged as potential biomarkers for poor microbiome health; these taxa were also enriched during VRE colonization and in the peri-LT compared with the pre-LT period across the patient cohort. The loss of protective taxa such as *F. prausnitzii*, Ruminococcaceae, Lachnospiraceae, and Bacteroidaceae, and an increase in pathogenic or inflammatory taxa[16,19], such as *Enterococcus* and Enterobacteriaceae, were also key signatures by using multiple differential abundance testing methods. We also identified post-transplant clinical complications that were independently associated with decreased microbiome diversity at peri- and post-LT phases, including

postoperative bleeding within 1 week of transplant, bile leak, and biliary stricture within the study period.

The robust antibiotic usage data available and high rates of antibiotic exposure in our cohort allowed us to interrogate the importance of specific antibiotic classes in shaping the gut microbiome. The observed reduction in α-diversity with recent antibacterial exposure was expected based on previous reports[14,15,39,40]. However, our ability to detect antibacterial class-dependent effects on gut α-diversity, most notably group 2 β-lactams and glyco-/lipo-peptides, is unique. Moreover, samples that were collected within 14 days of treatment with any antibiotic clustered separately, regardless of drug used or pre-/post-LT sampling time point, from samples not collected within this window, indicating that the gut community is still significantly altered even 2 weeks after the most recent antibiotic course. These findings have direct relevance for clinical use of specific antibacterial agents, given their impact on the gut microbiome, and for long-term complications and outcomes after LT, such as metabolic syndrome. Moreover, these dysbiotic changes due to antibiotic exposure were profound despite the high incidence of post-transplant complications.

In addition, the results reported here have important implications for clinical management in the liver and other solid organ transplantation. We identified distinct temporal signals for the relationship between colonization by specific MDRB and the gut microbiome. Before LT, both CRE and VRE colonization were predictors of reduced α-diversity. In the peri-LT period, Ceph-RE colonization was a significant predictor of lower α-diversity. CRE colonization again was significantly associated with early post-LT α-diversity in our multivariate models, as was VRE colonization in the late post-LT phase. These findings, in connection with our previous work[5], point to enrichment of CRE and VRE before LT in the setting of severe liver disease. The subsequent selection for Ceph-RE in the peri-LT period may be linked to the standard prophylactic antibiotic regimen prescribed (i.e., ampicillin/sulbactam), while frequent antibiotic use and complications likely led to the reemergence of CRE and VRE later in the post-transplant course. Preliminary evidence also suggests that the metabolism of tacrolimus is affected by certain gut microbiota, primarily Clostridiales, with potential ramifications on the variability of immunosuppression in transplant recipients[41]. Clinical decisions should therefore be made with specific considerations for microbial health to ensure long-term restoration of health after transplantation.

Without universal screening protocols, MDRB colonization is a silent predictor of subsequent infection[5]. However, we found that pre-transplant microbial dysbiosis is associated with increased likelihood of subsequent colonization by MDRB, which may inform preventive or interventional practices. For example, prevention of bacterial infections post-LT by using pre- or probiotics has previously been attempted[42–45], yet our results highlight the importance of clinical interventions aimed at preventing MDRB colonization as well as infection.

Recently, microbiome-targeted therapeutic approaches have become an area of significant interest, and have been used in limited clinical settings[42,44–46]. Probiotics or FMT, for example, may offer benefits to patients; however, they do not come without potential risks or side effects[47]. Identifying high-risk patients by microbial and/or clinical signatures will allow for tailored interventions with improved outcomes. Beyond the presence of MDRB, the need for peri-operative adjustments to antibiotic regimen, repeated exposures to class II β-lactams and/or glyco-/lipo-peptides, or the identification postoperative bleeding, bile leak, and/or biliary stricture could be useful markers to trigger microbiome interventions at specific pre- and post-transplant phases, as shown by our multivariate modeling.

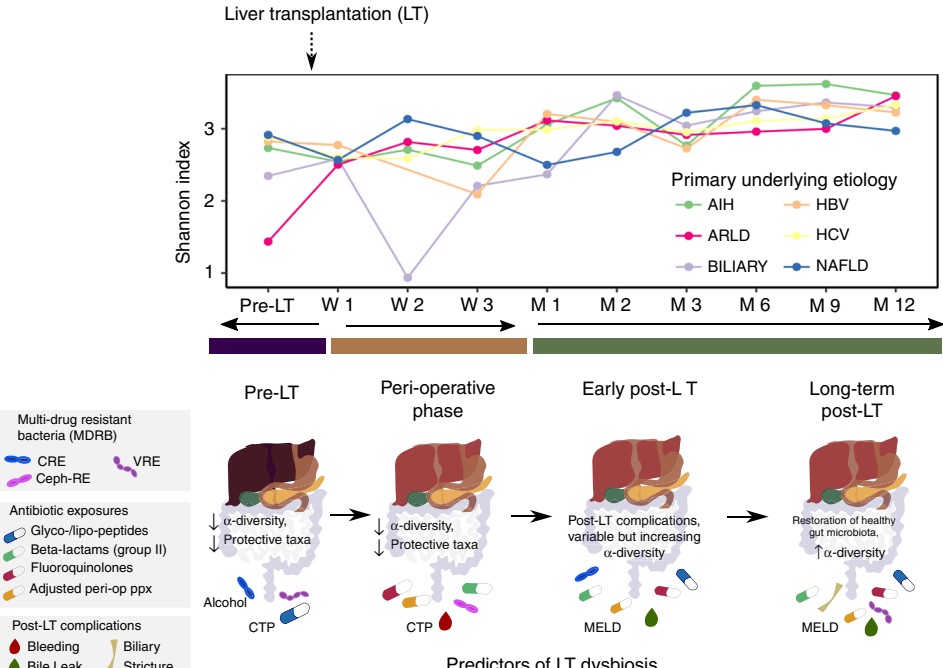

**Fig. 4** Microbial and clinical predictors of gut microbiome diversity across pre- and post-transplant phases. (Top) Top panel shows average Shannon α-diversity at each sampling time point within each disease category. (Left) Colonization by CRE or VRE, exposure to glyco-/lipo-proteins (i.e., vancomycin, daptomycin) within 14 days, diagnosis of ARLD, and high CTP score were independent predictors of decreased pre-transplant Shannon α-diversity (multivariate linear regression P < 0.0001). (Center) Peri-LT Shannon α-diversity was significantly lower given Ceph-RE colonization, exposure to group 2 β-lactams (e.g., piperacillin–tazobactam) or fluoroquinolones, high CTP score, peri-operative antibiotic regimen adjustments, and post-operative bleeding within 7 days of transplant (multivariate linear mixed-effect regression P < 0.0001). (Right) In the early post-LT period (months 1–3), CRE colonization was an independent predictor of Shannon α-diversity after adjusting for exposures to group 2 β-lactams, fluoroquinolones, and/or glyco-/lipo-proteins, high MELD, peri-operative antibiotic adjustment, and bile leak within 1 year post-LT (multivariate linear mixed-effect regression P < 0.0001). (Right) By the late post-LT phase (months 6–12), VRE colonization, group 2 β-lactam exposure, high MELD, peri-operative antibiotic adjustment, and bile leak or biliary stricture within 1 year post-LT were all independent predictors of Shannon α-diversity (multivariate linear mixed-effect regression P < 0.0001). LT liver transplant, AIH autoimmune hepatitis, ARLD alcohol-related liver disease, BILIARY biliary-related etiologies, HBV hepatitis B virus, HCV hepatitis C virus, NAFLD nonalcoholic fatty liver disease, MDRB multidrug-resistant bacteria, CRE carbapenem-resistant Enterobacteriaceae, VRE vancomycin-resistant enterococci, Ceph-RE Enterobacteriaceae resistant to third-generation cephalosporins, CTP Child–Turcotte–Pugh score, MELD model for end-stage liver disease. Source data are provided as a Source Data file

There are several limitations of this study. First, this was a single-center study, and LT cohorts, procedures, and outcomes may differ at other centers. Second, not all patients transplanted during the study period at our center participated; however, enrolled and non-enrolled patients did not differ in their clinical and demographic characteristics[5]. Third, our focus was the diversity and relative prevalence rather than absolute abundance of microbiota and may not address potential relationships between, for example, gut bacterial overgrowth and clinical outcomes. We did not have information on donor antibiotic exposures or MDRB colonization, though these variables are unlikely to impact post-transplant MDRB colonization. We do not have information on the birthplace or recent travel history of participants, although note that recent travel is limited in this population. Moreover, comparative genomics of CRE and Ceph-RE isolates demonstrated that these were closely related to clones circulating in our geographic region, making acquisition abroad less likely[5]. Last, not all patients provided pre-transplant samples, and not all patients contributed samples to all post-LT time points. Nevertheless, our data represent the largest LT cohort reported with extensive longitudinal follow-up period. The robustness of our cohort allowed for interrogation of multiple predictors/confounders in multivariate statistical models, including multiple liver disease etiologies and post-transplant complications, although we were likely unable to account for all potential confounders.

Our results highlight complex modulation of the gut microbiome by dominating bacteria, which are not only pathogenic but also drug-resistant. Beyond the direct impact on clinical decision-making in the case of LT, the growing importance of MDRB during and after transplantation of other solid organs makes our findings more broadly applicable. Ultimately, the use of microbial diversity and dysbiosis as a screening tool could allow clinicians to predict and ultimately mitigate or prevent complications, including those involving MDRB, during the post-transplant clinical course.

## Methods

**Patient recruitment and sample collection**. We prospectively recruited consecutive adult patients (age ≥ 18 years) undergoing LT at a tertiary care hospital between March 2014 and January 2017[5]. All participants provided informed consent. Fecal samples were collected at the time of enrollment, weekly during transplant hospitalization, and 2, 3, 6, 9, and 12 months post transplant (Supplementary Fig. 1) and stored at −80 °C. Sampling time points were aligned with standard postoperative clinic visit schedules at our center. We defined pre-transplant, peri-transplant (weeks 1–3), early post-transplant (months 1–3), and late post-transplant (months 6–12) sampling periods for subsequent analyses. Subjects without at least one sample were excluded. Study procedures were approved by the Columbia University Irving Medical Center Institutional Review Board (IRB-AAAM7704).

**Microbial and clinical data collection**. Stool samples were cultured on selective chromogenic agar (DRG International) to determine colonization with Ceph-Re, CRE, and/or VRE[5]. Patients were considered colonized by MDRB if their stool contained one or more MDRB (CRE-, VRE-, and/or Ceph-RE-positive) at a given time point. Inpatient antibiotic usage data up to 1 year post-LT were extracted from medical

records. Samples were coded based on receipt of at least one dose of a given antibiotic class (intravenous or oral) up to 14 days prior to sample collection to optimize identification of post-treatment alterations in microbial communities over a range of antibiotics[40] (see Supplementary Methods). Standard peri-operative (i.e., ampicillin/sulbactam) and post-transplant (i.e., trimethoprim/sulfamethoxazole) prophylaxis were not included in our antibiotic exposure analysis, as all patients were expected to receive standard regimens. However, we separately evaluated the impact of adjustments to peri-operative antibiotics, primarily due to antibiotic allergies or prior MDRB colonization and/or infection per discretion of the clinical team, compared to standard peri-operative antibiotic prophylaxis.

The data on clinical covariates were collected including demographics, co-morbidities, transplant indication, and liver disease severity at the time of LT (MELD[48], CTP score). The primary etiology of liver failure was classified as one of the following: HCV, NAFLD, ARLD, AIH, HBV, biliary causes (see Supplementary Methods), polycystic liver/kidney disease, or cryptogenic liver disease. Concurrent HCC was documented. For the surgical procedure, we collected ischemia time, donor type, and peri- and postoperative complications from enrollment until 1 year post transplant, including postoperative bleeding, biliary complications, and readmission to the hospital and/or ICU. In contrast to the complex liver–gut interactions found in patients with liver disease and cirrhosis, the transplant of healthy livers into a sick patient is unlikely to be influenced by dynamics of the donor gut. Therefore, we did not consider donor antibiotic exposure or MDRB colonization data in our modeling. Full details regarding patient, transplant, and outcome metadata coding are described in Supplementary Methods.

**DNA extraction, library preparation, and sequencing**. We extracted DNA from stool samples by using the MagAttract PowerSoil DNA Kit and amplified the 16S rRNA V3/V4 region by using established primers with Illumina Nextera adaptors[49]. Libraries were multiplexed by using Illumina Nextera XT Index kits, normalized, and pooled with 10% PhiX prior to sequencing on an Illumina MiSeq or HiSeq (Supplementary Methods). Sequencing data are publicly available through the NCBI Sequencing Read Archive (SRA) (accession number SRP185798 [https://www.ncbi.nlm.nih.gov/sra/?term=SRP185798]) after filtering any human-derived sequences.

**16S rRNA microbiome analysis**. 16S rRNA sequences were processed by using QIIME[50] and R v3.3.0[51]. Demultiplexed FASTQ sequences were quality-filtered, trimmed, de-replicated, and filtered for chimeric sequences by using QIIME through the National Institute of Allergy and Infectious Diseases Nephele platform[52]. Filtered sequences were clustered into operational taxonomic units (OTUs) at 97% similarity and aligned against the Greengenes 97% database for taxonomic classification. OTUs with an average relative abundance of <0.005% across all samples were filtered by using the *phyloseq* v1.19.1[53] package via Bioconductor[54] in R. Within-sample α-diversity (Shannon, Chao) and across-sample pairwise distance or β-diversity (Bray–Curtis, UniFrac, and weighted UniFrac) were calculated by using *phyloseq*. Based on α-diversity rarefaction, we applied a minimum cutoff of 7500 counts for inclusion in the analysis. We sequenced 723 fecal samples; 703 samples (median 4 per patient) passed the minimum count cutoff and were included in this analysis (Supplementary Fig. 1).

**Statistical testing of microbiome and clinical data**. Normal distribution of α-diversity metrics was confirmed by using the Shapiro test in R. Univariate linear regression models (*lm* in R) with α-diversity as the outcome were used for single-time-point analyses (e.g., pre-LT or week 1 diversity). Linear mixed-effect regression models (LME; *lme4* and *lmerTest* packages in R) were used when evaluating multiple time points to account for serial sampling (e.g., peri-LT or post-LT diversity). We first assessed univariate associations between microbial and clinical predictors relevant to the pre-, peri-, and early and late post-LT periods and microbiome α- and β-diversity. Predictors resulting in *P*-values ≤ 0.1 in univariate analysis were used to construct multivariate models BY using backward selection. We used either linear (pre-LT) or linear mixed-effect (peri- and early and late post-LT) regression to evaluate the ability of clinical and microbial factors to independently predict Shannon α-diversity during the following sampling periods: (1) pre-transplant; (2) peri-transplant (weeks 1–3); (3) early post-transplant (months 1–3); (4) late post-transplant (months 6–12). We compared the multivariate LME models with null models for each sampling period, including only random effects due to inter-patient variability, by using ANOVA to obtain overall *P*-values.

In order to assess temporal trends in mean α-diversity within each liver disease etiology population (Fig. 2), we used constrained linear mixed-effect (CLME) modeling by using the *CLME* package in R[55]. Principal coordinate analysis through *phyloseq* and permutational multivariate analysis of variance (PERMANOVA) tests through *vegan* v2.4–4[56] were used to test differences in microbial community composition (β-diversity) across groups based on pairwise UniFrac distance matrices. PERMANOVA tests the null hypothesis that the centroids and dispersion of objects across multiple groups are equivalent; therefore, significance indicates that objects cluster differently (different centroid and/or spread) across groups. *DESeq2* v1.14.1[57] was used to identify differentially abundant bacterial taxa, with significance defined as *P* < 0.05, and Benjamini–Hochberg-adjusted *P* (FDR) <0.05. Due to the potential for high false-positive rates by using *DESeq2* as described

previously[37], we also performed Analysis of Composition of Microbiomes (ANCOM)[38] by using the *ANCOM 2.0* package in R to refine our identification of bacterial taxa enriched across sample groups. All W-statistic cutoffs from ANCOM output (0.6, 0.7, 0.8, and 0.9) are reported in each supplementary table; for interpretation, significance was defined as W>0.7.

**Reporting summary**. Further information on research design is available in the Nature Research Reporting Summary linked to this article.

## Data availability

Sequencing data are publicly available through the NCBI Sequencing Read Archive (SRA) (accession number SRP185798) after filtering any human-derived sequences. All relevant code and metadata used for analyses included in this paper, and generation of Figs. 1–4, Supplementary Fig. 2, and Supplementary Tables 3–24 are published in a public repository at https://github.com/mka2136/lt_microbiome. The data underlying Figs. 1–4, Supplementary Fig. 2, and Supplementary Data 3–24 are provided as Source Data files. All other data are available from the corresponding author upon reasonable requests.

## Code availability

Custom scripts (R Markdown) and the required metadata used for analyses included in this paper are publicly available at https://github.com/mka2136/lt_microbiome.

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

## Acknowledgements

This study was funded through R01 AI116939 (to A.-C.U.) and 3R01 AI116939-02S1 and 5T32AI100852 (to A.G.-S.).

## Author contributions
This study was conceived by A.-C.U and designed by A.-C.U., M.K.A. and A.G.-S. S.B.S., S.D.K., M.J.G., S.S., G.K. and R.N. assisted in patient recruitment and sample collection and processing. A.G.-S., N.M., A.K., E.C.V. and A.-C.U. reviewed medical charts to collect the relevant metadata. M.K.A. analyzed the data, and M.K.A. and A.-C.U. interpreted the results with input from A.G.-S and E.C.V. M.K.A. wrote the initial paper draft, and revisions were made primarily in conjunction with A.-C.U. All authors provided feedback and approved the final paper.

## Competing interests
The authors declare no competing interests.
