## [Peer Review File · Nature Communications]

Reviewers' comments:

Reviewer #1 (Remarks to the Author):

Annavajhala et al. present a manuscript focusing on the evolution of the intestinal microbiome after liver transplantation (LT) and namely pre-LT microbial markers of subsequent colonization by MDRO. The central aim in this observational cohort study was to explore relationships between MDRO colonization and intestinal microbial dysbiosis pre- and post-LT. The major finding of this manuscript are that there is a strong association between microbiome diversity and intestinal colonization with MDRO and that this is dependent on the antibiotic class being used. This finding, along with others made in the paper, are novel and of interest to the biomedical community. This manuscript includes new information that may influence thinking in the field. The work is convincing but could benefit from more and clearer explanations of the methods, including the statistics, used to draw conclusions, therefore making the manuscript more accessible and potentially generalizable to readers of diverse backgrounds. Please see specific comments below, all of which are intended to strengthen the manuscript, and many of which should be easily addressable:

Abstract:

- Greater granularity is needed. For example, was there a hypothesis being tested, what was the setting (single-center study?), what was the primary endpoint, etc.

Introduction:

- The first time the concept of “reduced microbial diversity” is introduced, the authors should consider a description of what this means and the consequences of reduced diversity. There is a brief comment that reduced post-LT diversity is associated with acute cellular rejection and bloodstream infections, but no explanation of microbial diversity and what the benefits of diversity are.

Methods:

- Was there a reason for the timing of fecal sampling that was chosen? (weekly, then at 2,3,6,9,12-month increments?)

Microbial and clinical data collection

- Were any data collected about the LT donors, ie: their antibiotic use in the last year, and how that might have affected post-LT MDRO? If not, this should be mentioned as a limitation/cofounder.

- Biostatistician review is likely needed given the nature of the study and the inherent complexity of the requisite analyses.

Results:

- Similar to the comment above, are there any data on the patient characteristics that could serve as a confounding factor for their differences in MDRO? (Birth place, travel history etc), aside from what is in table S1?
- Would define the difference between alpha and beta diversity when first used, as this may not be well known to many in the LT field.
- How was it determined that gut community structure was altered in samples collected within 14 days of antibacterial treatment (and isn't this an expected finding)? The statistics portion should be better explained in simpler terms, or PERMANOVA should be explained earlier so it can be understood by a wider audience

Distinct gut microbiome dynamics in early and late post-operative phases

- page 9, 3rd sentence, "Colonization by Ceph-RE was an independent predictor of reduced peri-LT Shannon diversity ($p < 0.05$)" was " α " purposefully left out or intentional?

Discussion:

- "Liver transplantation" is spelled out in one part of the discussion and abbreviated as LT in an earlier part (as well as in earlier sections); please use abbreviation consistently.
- "More specifically, we identified pre-LT α -diversity as a significant marker for CRE colonization after transplantation." Can this sentence be more specific, either stating that a low or high α -diversity is as significant marker for CRE

Sentence 250, would consider using present tense, "However, our ability to detect antibacterial class-dependent effects on gut α -diversity, most notably group 2 beta-lactams and glyco-/lipo-peptides, was unique.

Tables

- No significant comments.

Figures:

- No significant comments.

Reviewer #2 (Remarks to the Author):

The relationship between multidrug-resistant organisms (MDRO) and gut dysbiosis in liver cirrhosis patients is not well understood and this article attempts to make an important contribution to this literature. They determine that MDRO colonization is a major marker of post liver transplant dysbiosis. While the results are potentially important, as noted below, I am concerned with the statistical methodology and interpretations of the data.

A. Figure 1: Although it is common for researchers to publish “busy” network plots, what exactly are we learning from these plots? If a dot is far from the clusters, what does it mean? Does it mean there is large variations in the distances? Only one edge is emerging from each node, why? Providing plots such as these without a deeper interpretation is not a useful exercise in my opinion.

B. Figure 2: The sample sizes are too small in many cases for the box plots to be useful. For example, what is the point in drawing a box plot when $n = 4$ or $n = 2$? In all such situations, statistical significance (and the p-values) of alpha, beta diversity is difficult to interpret.

C. Figure 3: What do the X-axis in panels A and B mean? What do the edges joining the circles mean? Are these network graphs of some kind? Are these plots results of unsupervised cluster analyses? I may have missed the details of how these plots were created and what they represent. I do see some clustering going on according to time. If the goal is to describe temporal dynamics of alpha and beta diversity, perhaps the authors may consider using trend analysis available through the freely available R based software package CLME. It allows for detecting mean trends over time in outcome variables such as alpha diversity after adjusting for covariates and accounting for repeated measurements. Panel C also requires some clarifications. Are these PCoA plots? Were these plots derived for each of the 6 groups of patients separately or did the authors perform a common PCoA analysis? The reason I ask is that for all 6 plots, the authors have same % for Axis 1 (14.4%) and Axis 2 (5.2%). More details are needed for these plots.

D. Differential abundance analysis in the supplementary text: Authors report a number of differential abundance analyses in the supplementary text. While, all differential abundance analyses described in the supplementary text are interesting and relevant for the study, I am fundamentally concerned with the statistical methodology used in the analyses. As reported in Figure 6 of Weiss et al. (Microbiome, 2017) DESEQ2 method (used in this paper) is subject to very large false discovery rates. In some simulations reported by Weiss et al. (2017), the FDR for DESEQ2

can be even as high as 70%. Thus, on average, 70% of the discoveries can be false. Consequently, I am not sure how to interpret all the significant taxa identified in the present paper.

Colonizing multidrug-resistant organisms and the longitudinal evolution of the intestinal microbiome after liver transplantation – Annavajhala et al.

Responses to Reviewers' comments

Reviewer #1 (Remarks to the Author):

Annavajhala et al. present a manuscript focusing on the evolution of the intestinal microbiome after liver transplantation (LT) and namely pre-LT microbial markers of subsequent colonization by MDRO. The central aim in this observational cohort study was to explore relationships between MDRO colonization and intestinal microbial dysbiosis pre- and post-LT. The major finding of this manuscript are that there is a strong association between microbiome diversity and intestinal colonization with MDRO and that this is dependent on the antibiotic class being used. This finding, along with others made in the paper, are novel and of interest to the biomedical community. This manuscript includes new information that may influence thinking in the field. The work is convincing but could benefit from more and clearer explanations of the methods, including the statistics, used to draw conclusions, therefore making the manuscript more accessible and potentially generalizable to readers of diverse backgrounds. Please see specific comments below, all of which are intended to strengthen the manuscript, and many of which should be easily addressable:

We thank the reviewer for their comments, and agree that the following clarifications will be helpful for many potential readers outside our field. Please see responses and edited text (highlighted) below.

Abstract:

- Greater granularity is needed. For example, was there a hypothesis being tested, what was the setting (single-center study?), what was the primary endpoint, etc.

To improve clarity and granularity of the abstract, we have added additional details to the abstract as suggested:

“We prospectively recruited 177 consecutive adult patients undergoing LT at a single-center tertiary care hospital over a three-year period and performed V3-V4 16S rRNA sequencing of 723 fecal samples, collected at defined intervals from pre-LT to one-year post-LT. We tested whether MDRO colonization was associated with decreased gut microbiome diversity before and after LT. In multivariate linear mixed-effect models, MDRO colonization predicted reduced Shannon α -diversity, after controlling for liver etiology and disease severity, antibiotic exposures, and clinical complications.” (Abstract)

Introduction:

- The first time the concept of “reduced microbial diversity” is introduced, the authors should consider a description of what this means and the consequences of reduced diversity. There is a brief comment that reduced post-LT diversity is associated with acute cellular rejection and bloodstream infections, but no explanation of microbial diversity and what the benefits of diversity are.

To clarify this point and add additional context to the concept of microbiome diversity, we have added additional details to the introduction:

“Microbiome diversity metrics quantify the number of taxa and their distribution within a sample (α -diversity) or relative similarity across samples in terms of overall microbial community composition (β -diversity). Increased diversity represents a healthy gut,³¹ as a varied commensal community may protect against overgrowth of pathogenic bacteria and help modulate the innate immune response at homeostasis.^{32,33} Mounting evidence is available regarding the association between low gut diversity and disease or poor clinical outcomes.³⁴ Dysbiosis of the gut, therefore, typically represents a significant perturbation in α - and/or β -diversity, and most generally is characterized by a reduction in gut microbiome diversity.” (Introduction, Lines 22-30)

Methods:

- Was there a reason for the timing of fecal sampling that was chosen? (weekly, then at 2,3,6,9,12-month increments?)

Sampling time-points were chosen to align with standard post-transplant clinic visit schedules at our center. We have added this explanation to the Methods section:

“Sampling time-points were chosen to align with standard post-operative clinic visit schedules at our center.” (Methods, Lines 409 – 410)

Microbial and clinical data collection

- Were any data collected about the LT donors, ie: their antibiotic use in the last year, and how that might have affected post-LT MDRO? If not, this should be mentioned as a limitation/cofounder.

While basic information regarding LT donors was available as part of the allocation process, accessing detailed donor information was not covered by our existing IRB. This would have required an additional consent process involving the living donor or deceased donor family at multiple institutions, posing significant logistical challenges. In addition, unlike FMT, for example, the transplant of “healthy” livers into a sick patient is unlikely to be influenced by dynamics of the donor gut, in contrast to the complex liver-gut interactions found in patients with liver disease and cirrhosis. For these reasons, we do not feel that detailed donor information should/need be included in our modeling.

“We did not have information on donor antibiotic exposures or MDRO colonization, though these variables are unlikely to impact post-transplant MDRO colonization.” (Discussion, Lines 387 – 390)

- Biostatistician review is likely needed given the nature of the study and the inherent complexity of the requisite analyses.

A biostatistical review was provided by reviewer 2. In response, we have added additional analyses, including Constrained Linear Mixed-Effect modeling (CLME) for longitudinal comparisons (Figure 3) and Analysis of Composition of Microbiomes (ANCOM) to augment our DESeq2-based differential abundance testing. These additional results can be found throughout the text and discussion.

Results:

- Similar to the comment above, are there any data on the patient characteristics that could serve as a confounding factor for their differences in MDRO? (Birth place, travel history etc), aside from what is in table S1?

Our group previously published a study which reported the incidence of post-LT MDRO colonization and infection as well as predictors of MDRO, from a subset of this LT cohort (Macesic et al., CID, 2018). We found that Child-Turcotte-Pugh score at the time of LT and duration of post-LT hospitalization were independent predictors of both MDR colonization and infection. We do not have information on the birth place or recent travel history of participants, although note that recent travel is limited in this population. We note that ethnicity and race were not significant predictors of MDRO colonization in the cohort. Our previous analyses also contained a detailed genomic characterization of CRE and ESBL colonizing and infectious isolates and demonstrated that these were closely related to other isolates from the same species frequently encountered in our geographic region.

- Would define the difference between alpha and beta diversity when first used, as this may not be well known to many in the LT field.

Thank you for this suggestion. We have added a brief introduction to the concepts of α - and β -diversity in the first section of the Results for those outside the microbiome field.

"We first analyzed within-sample α -diversity, which incorporates the richness (number of taxa) and evenness (how these taxa are distributed) within the community. In unadjusted linear mixed-effect regression analysis, CRE, VRE, and Ceph-RE colonization were each associated with reduction in both Shannon and Chao α -diversity across all time-points (Shannon $p < 0.0001$; Chao $p < 0.01$, **Table 2, Table S1**). VRE-colonized (culture-positive) compared to VRE-negative samples showed the most drastic reduction in both diversity indices (mean Shannon of 2.32 ± 0.86 vs. 3.08 ± 0.73 and Chao of 189 ± 73 vs. 250 ± 75 , respectively).

Microbiome β -diversity compares microbial communities between samples by generating a dissimilarity matrix, with pairwise distances calculated for every pair of samples. The permutational multivariate analysis of variance (PERMANOVA) test can then be used to assess whether overall microbiome community structure (presence and relatedness of taxa across samples) is significantly different across two groups based on the β -diversity distance matrix." (**Results**, Lines 59-71)

- How was it determined that gut community structure was altered in samples collected within 14 days of antibacterial treatment (and isn't this an expected finding)? The statistics portion should be better explained in simpler terms, or PERMANOVA should be explained earlier so it can be understood by a wider audience

While we expected antibiotics to eradicate a certain proportion of bacteria in the gut, the results reported here quantify the selection for/against specific taxa, resulting in a shift in community structure (determined by PERMANOVA, described in more detail below).

This indicated that samples that were collected within 14 days of treatment with any antibiotic clustered separately, regardless of drug used or pre-/post-LT sampling time-point, from samples not collected within this window. Furthermore, prior reports generally did not report this long of an antibiotic-sampling interval; what we show here points out that even 2 weeks after the last antibiotic course, the gut community is still significantly altered. We do note that it will be important in future studies to provide a more in-depth and more granular analysis of the impact of specific antibiotics on the gut microbiome, both in similar and other cohorts.

We have updated the Results and Methods sections to include a more generalized description of the PERMANOVA test to aid in interpretation of the results by a broader audience.

“Microbiome β -diversity compares microbial communities between samples by generating a dissimilarity matrix, with pairwise distances calculated for every pair of samples. The permutational multivariate analysis of variance (PERMANOVA) test can then be used to assess whether overall microbiome community structure (presence and relatedness of taxa across samples) is significantly different across two groups based on the β -diversity distance matrix.” (Results, Lines 66-70)

“Principle coordinate analysis through *phyloseq* and permutational multivariate analysis of variance (PERMANOVA) tests through *vegan* v2.4-4⁵⁰ were used to test differences in microbial community composition (β -diversity) across groups based on pairwise UniFrac distance matrices. PERMANOVA tests the null hypothesis that the centroids and dispersion of objects across multiple groups are equivalent; therefore, significance indicates that objects cluster differently (different centroid and/or spread) across groups.” (Methods, Lines 479 – 484)

Distinct gut microbiome dynamics in early and late post-operative phases

- page 9, 3rd sentence, “Colonization by Ceph-RE was an independent predictor of reduced peri-LT Shannon diversity ($p < 0.05$)” was “ α ” purposefully left out or intentional?

This omission was an oversight; we have now added the Greek alpha as shown below:

“Colonization by Ceph-RE was an independent predictor of reduced **peri-LT** Shannon α -diversity ($p < 0.05$) after adjusting for exposure to group 2 β -lactams or fluoroquinolones, high CTP score, peri-operative antibiotic regimen adjustment, and post-operative bleeding (multivariate LME model $p < 0.0001$) (Figure 4B).” (Results, Lines 224 – 230)

Discussion:

- “Liver transplantation” is spelled out in one part of the discussion and abbreviated as LT in an earlier part (as well as in earlier sections); please use abbreviation consistently.

Thank you for bringing this to our attention; we have edited the text accordingly to use “LT” throughout after the first definition of the abbreviation.

- “More specifically, we identified pre-LT α -diversity as a significant marker for CRE colonization after transplantation.” \diamond Can this sentence be more specific, either stating that a low or high α -diversity is as significant marker for CRE

We have added this detail regarding directionality of the association between pre-LT α -diversity and post-LT CRE colonization.

“More specifically, we identified **low** pre-LT α -diversity as a significant marker for CRE colonization after transplantation.” (**Discussion**, Lines 313 – 315)

Sentence 250, would consider using present tense, “However, our ability to detect antibacterial class-dependent effects on gut α -diversity, most notably group 2 beta-lactams and glyco-/lipo-peptides, was unique.

We have edited this sentence accordingly:

“However, our ability to detect antibacterial class-dependent effects on gut α -diversity, most notably group 2 beta-lactams and glyco-/lipo-peptides, **is** unique.” (**Discussion**, Lines 342 – 344)

Tables

- No significant comments.

Figures:

- No significant comments.

Reviewer #2 (Remarks to the Author):

The relationship between multidrug-resistant organisms (MDRO) and gut dysbiosis in liver cirrhosis patients is not well understood and this article attempts to make an important contribution to this literature. They determine that MDRO colonization is a major marker of post liver transplant dysbiosis. While the results are potentially important, as noted below, I am concerned with the statistical methodology and interpretations of the data.

We thank the reviewer for their thorough review of our statistical methods, and appreciate the suggestions and comments below. We have updated the manuscript by applying the suggested analyses, including the CLME R package for longitudinal analysis of α -diversity outcomes and ANCOM as a robust differential analysis approach to augment our DESeq2 results. Both the text and figures (Fig. 1-3) have been updated accordingly. Please see responses and edited text (highlighted) below.

A. Figure 1: Although it is common for researchers to publish “busy” network plots, what exactly are we learning from these plots? If a dot is far from the clusters, what does it mean? Does it mean there is large variations in the distances? Only one edge is emerging from each node, why? Providing plots such as these without a deeper interpretation is not a useful exercise in my opinion.

We appreciate the concern about the busy network plots in Figure 1. These were intended to show that, based on Bray-Curtis β -diversity, samples from which we were able to culture MDRO (CRE, VRE, Ceph-RE, or any of the three) clustered separately from samples with none of these MDRO. In response to your comment, however, we have now revised Figure 1 to address this concern. First, we used UniFrac β -diversity rather than Bray-Curtis to be more consistent with other figures (Figures 2, 3) and tables (Table 3). Second, rather than a series of network plots, we performed non-metric multidimensional scaling (NMDS) ordination of the UniFrac distance matrix for the full dataset of 703 samples. The presence/absence (based on culture) of CRE, VRE, Ceph-RE, or any of the above in each sample was then overlaid in color on the resulting NMDS plot. We ran PERMANOVA tests to verify the observed clustering by MDRO colonization; samples colonized by CRE, VRE, Ceph-RE, and any of the three significantly clustered together, though the pseudo-F-statistic values for each show slight differences in the effects of the different MDRO types on overall microbiome community structure. This information is reflected in the updated Figure 1, Figure 1 legend, and Table 3.

“Figure 1. Non-metric multidimensional scaling (NMDS) of UniFrac β -diversity reveals significant clustering of stool samples collected during colonization by multidrug-resistant organisms (MDRO). NMDS plots were constructed using UniFrac β -diversity. Each point represents a single sample in the full dataset (n=703). The difference in microbiome community structure was compared during vs. not during colonization by (A) any MDRO (dark yellow; colonized n=254 vs. non-colonized n=449), (B) vancomycin-resistant enterococci (VRE, dark pink; colonized n=142 vs. non-colonized n=561), (C) Enterobacteriaceae resistant to third-generation cephalosporins (Ceph-RE, dark green; colonized n=162 vs. non-colonized n=541), and (D) carbapenem-resistant

Enterobacteriaceae (CRE, dark purple; colonized n=46 vs. non-colonized n=657) as identified using culture-based methods (see **Methods, Supplemental Methods**). PERMANOVA pseudo-F-statistic and P-value are shown.” (Legend, Figure 3)

B. Figure 2: The sample sizes are too small in many cases for the box plots to be useful. For example, what is the point in drawing a box plot when n= 4 or n= 2? In all such situations, statistical significance (and the p-values) of alpha, beta diversity is difficult to interpret.

We agree with the reviewer on this point; for the alpha-diversity plot by primary underlying cause of liver disease (Figure 2A), we have edited the plot remove the low-frequency “BILIARY” (n=4) and “PCLD” (n=2) categories. We do want to note that these two categories were not included in the multivariate modeling in subsequent sections of the manuscript, and therefore would not affect those models or interpretations. The PERMANOVA of the beta-diversity by primary indication (Figure 2B) was based on individual binary variables (Yes/No for autoimmune hepatitis (AIH) and Yes/No for alcohol-related disease (ARLD) were independently assessed); therefore removing or keeping these low-frequency indications do not affect the outcome. The legend for Figure 2 has been updated to reflect these additional details/changes as follows:

“**Figure 2. Pre-transplant α - and β -diversity associated with liver disease etiology and severity.** (A) Shannon and Chao α - and (B) UniFrac β -diversity indices stratified by primary indication for LT (AIH n=7; ARLD n=7; HBV n=8; HCV n=37; NAFLD n=14). (C, D) α - and UniFrac β -diversity stratified by high vs. low MELD score at time of LT (> 18 (n=43) vs. \leq 18 (n=40)). (E, F) α - and UniFrac β -diversity stratified by CTP class at time of LT (A (n=16) vs. B (n=24) vs. C (n=43)). (A, C, E) α -diversity plots show p-values for univariate linear regression (see Table 2) as follows: + ($p < 0.1$); * ($p < 0.05$); ** ($p < 0.01$); *** ($p < 0.001$); **** ($p < 0.0001$). (B, D, F) PERMANOVA P-values calculated using UniFrac β -diversity distances are shown. **Abbreviations:** LT – liver transplant; AIH – autoimmune hepatitis; ARLD – alcohol-related liver disease; BILIARY – biliary-related etiologies; HBV – hepatitis B virus; HCV – hepatitis C virus; NAFLD – non-alcoholic fatty liver disease; PCLD – polycystic liver/kidney disease; MELD – model for end-stage liver disease; CTP – Child-Turcotte-Pugh.” (Legend, Figure 2)

C. Figure 3: What do the X-axis in panels A and B mean? What do the edges joining the circles mean? Are these network graphs of some kind? Are these plots results of unsupervised cluster analyses? I may have missed the details of how these plots were created and what they represent. I do see some clustering going on according to time. If the goal is to describe temporal dynamics of alpha and beta diversity, perhaps the authors may consider using trend analysis available through the freely available R based software package CLME. It allows for detecting mean trends over time in outcome variables such as alpha diversity after adjusting for covariates and accounting for repeated measurements.

We apologize for this oversight on our part; the x-axes for panels A and B were not labeled, and the legend was unclear in how these were generated. These two panels are meant to show each α -diversity metric (A: Shannon; B: Chao, y-axis) over time (days

post-LT, x-axis); samples from individual patient are connected by a gray line. As the reviewer pointed out, these plots were meant to study the temporal dynamics of α -diversity over time for each disease group. We have now updated our analyses using the CLME package as suggested and were able to show the impact of sampling time-point on both Shannon and Chao α -diversity was significant for most of the six major diagnoses, except HBV, and we were able to define which time intervals were most critical in terms of microbiome response (change in diversity). These changes are now reflected in the updated Figure 3, the Figure 3 legend, and in the results, discussion, and methods sections as described below. We have also added supplemental tables Table S9 and Table S10 with the outputs from the CLME analyses:

“Figure 3. Progression of the gut microbiome in various liver disease populations before and after transplantation. Shannon (A) and Chao (B) α -diversity and UniFrac β -diversity (C) throughout the study period are shown. (A, B) AIH (13 patients, 45 samples), ARLD (19 patients, 69 samples), BILIARY (18 patients, 58 samples), HBV (10 patients, 45 samples), HCV (71 patients, 304 samples), and NAFLD (30 patients, 115 samples) were the most represented indications for LT in our cohort. These panels show Shannon (A) and Chao (B) α -diversity within each sample by the number of days post-LT of sampling, for each disease group. Constrained linear mixed-effect (CLME) analysis was used to determine a global P-value describing the temporal trend in α -diversity within each disease group (see Tables S9 and S10).” (Legend, Figure 3)

“We used constrained linear mixed-effect (CLME) analysis to test for significant changes in α -diversity over time within each disease group (Table S9, Table S10). Based on global bootstrap LRT testing through CLME, both Shannon and Chao α -diversity significantly differed across time-points for almost all major diagnoses (AIH, ARLD, biliary-related disease, and HCV; CLME $p < 0.05$). Only Chao α -diversity showed a significant longitudinal progression in patients with NAFLD (Shannon $p = 0.140$; Chao $p < 0.05$), and HBV patients showed no significant temporal trend in either α -diversity metric. Of note, for patients with ARLD, both Shannon and Chao α -diversity were lowest pre-LT, while for all other diagnoses, Week 1 post-LT represented the most significant reduction in α -diversity. More granular comparisons across time-points using CLME (individual θ and p-values) revealed interesting trends in α -diversity over time: for patients with AIH, biliary-related etiologies, and HCV, increases in α -diversity were significant ($p < 0.05$) primarily between Week 3 to Month 1 post-LT and Month 3 to Month 6 post-LT. This further justified our decision to analyze the early (Month 1-3) and late (Month 6-12) post-LT phases separately in subsequent analyses. In contrast, ARLD patients, likely due to low pre-LT diversity, had a drastic increase in Shannon and Chao α -diversity earlier in their post-LT course (pre-LT to Week 2 post-LT).” (Results, Lines 154-177)

“In order to assess temporal trends in mean α -diversity within each liver disease etiology population (Figure 2), we used constrained linear mixed-effect (CLME) modeling using the CLME package in R.⁴⁹” (Methods, Lines 477 – 479)

Panel C also requires some clarifications. Are these PCoA plots? Were these plots derived for each of the 6 groups of patients separately or did the authors perform a common PCoA analysis? The reason I ask is that for all 6 plots, the authors have same % for Axis 1 (14.4%) and Axis 2 (5.2%). More details are needed for these plots.

Please see the updated legend for Figure 3, which was updated to include more details about all three panels. The reviewer is correct that these plots were not derived independently; rather, all points were plotted on a single PCoA plot and then separated into the panels by diagnosis for easier visual inspection of the temporal trends, as our primary question was the shift in community structure over time within a disease group.

“(C) Changes in UniFrac β -diversity in the pre-, peri- (Weeks 1-3) and early (Months 1-3) and late (Months 6-12) post-transplant period are shown for patients with AIH (pre-LT n=7; peri-LT n=17; early post-LT n=10; late post-LT n=11), ARLD (pre-LT n=7; peri-LT n=20; early post-LT n=20; late post-LT n=22), BILIARY (pre-LT n=4; peri-LT n=17; early post-LT n=12; late post-LT n=25), HBV (pre-LT n=8; peri-LT n=11; early post-LT n=11; late post-LT n=15), HCV (pre-LT n=37; peri-LT n=88; early post-LT n=89; late post-LT n=90), and NAFLD (pre-LT n=14; peri-LT n=38; early post-LT n=31; late post-LT n=32). UniFrac β -diversity was calculated and principal coordinates analysis (PCoA) was performed with the full data-set, and samples from each disease population were subsequently separated for visualization of temporal trends. Across all samples, PCoA axis 1 explained 14.5% of observed variance, and PCoA axis 2 explained 5.6% of observed variance.” (Legend, Figure 3)

D. Differential abundance analysis in the supplementary text: Authors report a number of differential abundance analyses in the supplementary text. While, all differential abundance analyses described in the supplementary text are interesting and relevant for the study, I am fundamentally concerned with the statistical methodology used in the analyses. As reported in Figure 6 of Weiss et al. (Microbiome, 2017) DESEQ2 method (used in this paper) is subject to very large false discovery rates. In some simulations reported by Weiss et al. (2017), the FDR for DESEQ2 can be even as high as 70%. Thus, on average, 70% of the discoveries can be false. Consequently, I am not sure how to interpret all the significant taxa identified in the present paper.

As suggested by the reviewer and by the findings from the Weiss et al. paper from 2017, we have updated our analyses to include the ANCOM approach as described below, to augment the DESeq2 results and validate our DESeq2-based findings with the more robust ANCOM methodology. For the majority of comparisons, ANCOM identified differentially abundant OTUs that were also identified by DESeq2, though with greater discriminatory power. This led to several updates in the results and discussion sections of our manuscript, as shown below:

“DESeq2 v1.14.151 was used to identify differentially abundant bacterial taxa, with significance defined as $p < 0.05$ and Benjamini-Hochberg adjusted p (FDR) < 0.05 . Due to the potential for high false-positive rates using DESeq2 as described previously,⁵² we also performed Analysis of Composition of Microbiomes (ANCOM)⁵³ using the ANCOM 2.0 package in R to refine our identification of bacterial taxa enriched across sample groups. All W-statistic

cutoffs from ANCOM output (0.6, 0.7, 0.8, and 0.9) are reported in each supplemental table; for interpretation, significance was defined as $W > 0.7$.” (Methods, Lines 486 – 491)

“Given the potential for high false-positive rates using *DESeq2*,³⁷ however, we also used the Analysis of Composition of Microbiomes (ANCOM) method to refine our identification of bacterial taxa enriched across group comparisons.³⁸ Using ANCOM, we confirmed that several *Streptococcus* spp. and *Lactobacillus* spp. OTUs, and *L. zaeae*, were differentially enriched pre-LT in patients with ARLD (W -statistic > 0.7) (Table S4).” (Results, Lines 112 – 117)

“*DESeq2* identified significant enrichment in *Enterococcus casseliflavus*, *Veillonella dispar*, and *L. zaeae* in high-MELD patients, and differential enrichment of *F. prausnitzii*, *Dialister* spp., *Bifidobacterium bifidum*, *Bifidobacterium adolescentis*, and multiple *Bacteroides* OTUs in low-MELD patients (FDR-adjusted $p < 0.05$) (Table S5). ANCOM analysis identified significant enrichment of *Enterococcus* and *Lactobacillus zaeae* in high-MELD patients and of taxa traditionally associated with a healthier gut (Lachnospiraceae, *F. prausnitzii*, *Collinsella aerofaciens*) in low-MELD patients ($W > 0.7$; Table S6).” (Results, Lines 121 – 134)

“CTP class C was associated with significant enrichment in several *Enterococcus* and *Lactobacillus* OTUs, while class A was characterized by enrichment in *F. prausnitzii*, *Bifidobacterium*, similarly to patients with low-MELD (*DESeq2*, FDR-adjusted $p < 0.05$) (Table S7). Notably, the same *Enterococcus* and *F. prausnitzii* OTUs were enriched pre-LT in high-MELD and CTP class C and low-MELD and CTP class A patients, respectively, based on both *DESeq2* and ANCOM (Table S5, Table S6, Table S7, Table S8).” (Results, Lines 137 – 142)

“We also used ANCOM to analyze the differential abundance of gut microbiota across transplant phases while adjusting for repeated sampling and liver disease etiology. As with *DESeq2*, both “pathogenic” taxa – e.g. *Enterococcus* (including *E. casseliflavus*), *Lactobacillus* (including *L. zaeae*), and *Streptococcus* – and “protective” taxa – e.g. *F. prausnitzii*, Ruminococcaceae – were differentially abundant across pre- to post-LT phases (Table S14).” (Results, Lines 208 – 213)

“Both *DESeq2* and ANCOM identified the same *Enterococcus* OTU as enriched pre-LT in patients who developed MDRO within 1-year post-LT, and ANCOM identified both *Blautia obeum* and *Dorea formicigenerans* as potential pre-LT markers of subsequent MDRO colonization (Table S16).” (Results, Lines 248 – 251)

“Overall, as expected based on culture results, samples collected during CRE colonization were differentially enriched in several Enterobacteriaceae OTUs, including *Klebsiella* spp. and *Enterobacter cloacae*, through both *DESeq2* and ANCOM analyses (Table S17, Table S18). These samples also had significantly

higher levels of *Streptococcus*, including *Streptococcus anginosus*. CRE-colonized samples harbored significantly reduced levels (*DESeq* FDR-adjusted $p < 0.05$, ANCOM $W > 0.7$) of *R. gnavus*, *F. prausnitzii*, and OTUs assigned to the *Bacteroides* and *Blautia* genera. Similarly, both *DESeq2* and ANCOM identified enrichment of several Enterobacteriaceae, including OTUs assigned to *Klebsiella* and *E. cloacae*, *Streptococcus*, and *Enterococcus* (including *E. casseliflavus*), and significantly lower levels of *Blautia*, *C. aerofaciens*, and several *Bacteroides* OTUs during colonization by Ceph-RE (Table S19, Table S20). VRE colonization corresponded with a significant (*DESeq* FDR-adjusted $p < 0.05$, ANCOM $W > 0.7$) decrease in *F. prausnitzii*, multiple Lachnospiraceae, and *Blautia* (Table S21, Table S22). Conversely, samples collected during VRE colonization were significantly enriched not only in *Enterococcus*, including *E. casseliflavus*, but also *L. zae*, *Staphylococcus* and *Klebsiella*.” (Results, Lines 252 – 267)

“Differential abundance testing using both *DESeq2* and ANCOM revealed that, across all time-points, those who were colonized by MDRO at any point were significantly enriched in *Enterococcus* (including *E. casseliflavus*), *Klebsiella*, *E. cloacae*, as well as *L. zae*, while those who never developed MDRO colonization were enriched in *Bacteroides*, *Blautia* (including *B. obeum*), *C. aerofaciens*, and multiple Lachnospiraceae and Ruminococcaceae OTUs (Table S23, Table S24).” (Results, Lines 272 – 277)

“Patients who developed MDRO colonization within one-year post-LT had distinct β -diversity compared to those who never became colonized, with increased *Enterococcus* and *Klebsiella* and reduced levels of *Bacteroides* and *Lachnospira* throughout the entire study period. Importantly, pre-LT microbial β -diversity and levels of specific taxa were also significantly altered in patients who subsequently developed MDRO colonization.” (Discussion, Lines 309 – 313)

REVIEWERS' COMMENTS:

Reviewer #1 (Remarks to the Author):

The authors responded appropriately to the concerns outlined. We only have a few remaining comments, as outlined, below:

1. Microbial and clinical data collection

- Were any data collected about the LT donors, ie: their antibiotic use in the last year, and how that might have affected post-LT MDRO? If not, this should be mentioned as a limitation/cofounder.
- The explanation given in blue makes a lot of sense. Would consider adding your explanation as to why those variables are unlikely to impact post-transplant MDRO colonization to the manuscript, possible example "The transplant of "healthy" livers into a sick patient is unlikely to be influenced by dynamics of the donor gut, in contrast to the complex liver-gut interactions found in patients with liver disease and cirrhosis. For these reasons, we did not collect information on donor antibiotic exposure or MDRO colonization.

2. Results:

- Similar to the comment above, are there any data on the patient characteristics that could serve as a confounding factor for their differences in MDRO? (Birth place, travel history etc), aside from what is in table S1?

Can consider referencing your previously published study and include a brief description of the below:

Our group previously published a study which reported the incidence of post-LT MDRO colonization and infection as well as predictors of MDRO, from a subset of this LT cohort (Macesic et al., CID, 2018). We found that Child-Turcotte-Pugh score at the time of LT and duration of post-LT hospitalization were independent predictors of both MDR colonization and infection. We do not have information on the birth place or recent travel history of participants, although note that recent travel is limited in this population. We note that ethnicity and race were not significant predictors of MDRO colonization in the cohort. Our previous analyses also contained a detailed genomic characterization of CRE and ESBL colonizing and infectious isolates and demonstrated that these were closely related to other isolates from the same species frequently encountered in our geographic region.

3. How was it determined that gut community structure was altered in samples collected within 14 days of antibacterial treatment (and isn't this an expected finding?)? The statistics portion should be better explained in simpler terms, or PERMANOVA should be explained earlier so it can be understood by a wider audience

The explanation in blue makes a lot of sense, would consider using more direct and simple language in the actual manuscript as well, consider adding this section to the manuscript:

“the results reported here quantify the selection for/against specific taxa, resulting in a shift in community structure (determined by PERMANOVA, described in more detail below). This indicated that samples that were collected within 14 days of treatment with any antibiotic clustered separately, regardless of drug used or pre-/post-LT sampling time-point, from samples not collected within this window. Furthermore, prior reports generally did not report this long of an antibiotic-sampling interval; what we show here points out that even 2 weeks after the last antibiotic course, the gut community is still significantly altered. We do note that it will be important in future studies to provide a more in-depth and more granular analysis of the impact of specific antibiotics on the gut microbiome, both in similar and other cohorts.”

Reviewer #2 (Remarks to the Author):

All of my concerns are addressed. I have no further concerns with the biostatistical methods and interpretation of the data.

Responses to Reviewer Comments

Reviewer #1 (Remarks to the Author):

The authors responded appropriately to the concerns outlined. We only have a few remaining comments, as outlined, below:

We thank the reviewer for their suggestions, and have incorporated many of our responses into the manuscript text, as detailed below.

1. Microbial and clinical data collection

- Were any data collected about the LT donors, ie: their antibiotic use in the last year, and how that might have affected post-LT MDRO? If not, this should be mentioned as a limitation/cofounder.
- The explanation given in blue makes a lot of sense. Would consider adding your explanation as to why those variables are unlikely to impact post-transplant MDRO colonization to the manuscript, possible example “The transplant of “healthy” livers into a sick patient is unlikely to be influenced by dynamics of the donor gut, in contrast to the complex liver-gut interactions found in patients with liver disease and cirrhosis. For these reasons, we did not collect information on donor antibiotic exposure or MDRO colonization.

We have included this explanation into this section under Methods:

“In contrast to the complex liver-gut interactions found in patients with liver disease and cirrhosis, the transplant of healthy livers into a sick patient is unlikely to be influenced by dynamics of the donor gut. Therefore, we did not consider donor antibiotic exposure or MDRO colonization data in our modeling.”

(Methods, Lines 573 – 576)

2. Results:

- Similar to the comment above, are there any data on the patient characteristics that could serve as a confounding factor for their differences in MDRO? (Birth place, travel history etc), aside from what is in table S1?

Can consider referencing your previously published study and include a brief description of the below:

Our group previously published a study which reported the incidence of post-LT MDRO colonization and infection as well as predictors of MDRO, from a subset of this LT cohort (Macesic et al., CID, 2018). We found that Child-Turcotte-Pugh score at the time of LT and duration of post-LT hospitalization were independent predictors of both MDR colonization and infection. We do not have information on the birth place or recent travel history of participants, although note that recent travel is limited in this population. We note that ethnicity and race were not significant predictors of MDRO colonization in the cohort. Our previous analyses also contained a detailed genomic characterization of CRE and ESBL colonizing and infectious isolates and demonstrated that these were closely related to other isolates from the same species frequently encountered in our geographic region.

Similarly, we have included a shortened version of this explanation into the Discussion section:

“We do not have information on the birth-place or recent travel history of participants, although note that recent travel is limited in this population.”

Moreover, comparative genomics of CRE and Ceph-RE isolates demonstrated that these were closely related to clones circulating in our geographic region, making acquisition abroad less likely.⁵ (Discussion, 515-519)

3. How was it determined that gut community structure was altered in samples collected within 14 days of antibacterial treatment (and isn't this an expected finding)? The statistics portion should be better explained in simpler terms, or PERMANOVA should be explained earlier so it can be understood by a wider audience

The explanation in blue makes a lot of sense, would consider using more direct and simple language in the actual manuscript as well, consider adding this section to the manuscript: "the results reported here quantify the selection for/against specific taxa, resulting in a shift in community structure (determined by PERMANOVA, described in more detail below). This indicated that samples that were collected within 14 days of treatment with any antibiotic clustered separately, regardless of drug used or pre-/post-LT sampling time-point, from samples not collected within this window. Furthermore, prior reports generally did not report this long of an antibiotic-sampling interval; what we show here points out that even 2 weeks after the last antibiotic course, the gut community is still significantly altered. We do note that it will be important in future studies to provide a more in-depth and more granular analysis of the impact of specific antibiotics on the gut microbiome, both in similar and other cohorts."

We have included the concepts above into both the Results and Discussion:

"Gut community structure was also significantly altered in samples collected within 14 days of treatment with any antibiotic, regardless of drug used or pre-/post-LT sampling time-point, compared to samples not collected within this window (UniFrac PERMANOVA $P=0.001$) (Table 4)." (Results, Lines 107-110)

"Moreover, samples that were collected within 14 days of treatment with any antibiotic clustered separately, regardless of drug used or pre-/post-LT sampling time-point, from samples not collected within this window, indicating that the gut community is still significantly altered even two weeks after the most recent antibiotic course." (Discussion, Lines 462-465)